



# Characteristics of a Hailstorm over the Andean La Paz Valley

Marcelo Zamuriano[1], Andrey Martynov[1], Luca Panziera[1,2], and Stefan Brönnimann[1]

[1]Oeschger Centre for Climate Change Research and Institute of Geography, University of Bern, Switzerland
[2]MeteoSwiss, Locarno Monti, Switzerland

**Correspondence:** Marcelo Zamuriano (marcelo.zamuriano@giub.unibe.ch)

**Abstract.** The iconic hailstorm and flash flood episode of 19 February 2002 over La Paz city is numerically investigated in this article. Large scale atmospheric circulation is dynamically downscaled in order to take into account the complex orography forcing and local features. Satellite observations suggests late morning shallow convection over the Altiplano that becomes deep convection in the early afternoon around complex orography. The control simulation captures well the cloud evolution and suggest a two-stage precipitation mechanism. First, early convection occurred around 1200 LST and originated from thermodynamic instability combined with lake breeze and orographic lifting. Rainfall discharge then generated cold pools. During the second stage, cold pools around complex orography were propagated by lake breeze and encountered the La Paz Valley breeze, triggering the deep convection near La Paz city around 1400 LST. We assess the importance of local features through numerical experiments, which include modification of orography, suppression of surface heat fluxes, changes of surface lake temperature and removal of the lake. We show the importance of orographic configuration as triggering mechanism for convection initiation and for mesoscale circulation, the role of lake temperature for frontal breeze and propagation of cold pools, and of surface heat fluxes for atmospheric instability. This study highlights the complex interaction between lakes, surface heating and orography that favour deep convection and hailstorm formation, which is especially relevant around the Titicaca lake region.

## 1 Introduction

Thunderstorms accompanied by hail episodes cause strong damage to the infrastructure, agriculture and people. In the Central Andes region, this kind of events is relatively frequent during austral summer and affects mainly the crops in rural parts and daily activities in the city. Over the Altiplano is hard to obtain a formal hail climatology from measurements. Nevertheless, some efforts have been done by taking into consideration Bolivian farmers' perception of extreme events over the Andes. Andean farmers perceive an increase of the frequency and intensity of storms and hail causing crop damage, especially when potatoes are in flower (Boillat and Berkes, 2013; Yasukawa, 2011).

As a glimpse of hail effects on urban parts, we take the 19 February 2002's episode in La Paz. It was the most iconic natural disaster related to hail and flash-flood known as "Black February". The local media and the Bolivian National Weather Service (SENAMHI) reported that on between 1420 and 1545 LST (local solar time) a hailstorm affected the city of La Paz. It caused such damage, both to individuals and infrastructure, that it was described as an unprecedented crisis. The event was thus presented as an exceptional natural disaster (Hardy, 2011; Yasukawa, 2011); and many testimonies, videos and pictures of



this iconic hailstorm can be found on-line. This natural disaster (that resulted in 69 casualties and still keeps in people's minds) exposed the city structural vulnerability, providing valuable but at the same time hard lessons to the local crisis managers.

Even if the impacts of hail are well known over the Altiplano region, there is no systematic official hail observation system; neither ground observations nor remotely by radar. Furthermore, no data from insurance companies on hail are available. Most
of the knowledge is taken from local people's perception of hail frequency and intensity. The lack of formal physical process knowledge about local thunderstorms formation over this region is evident as we take as example the explanation given by the SENAMHI about the plausible mechanisms for this particular cell formation.

The SENAMHI argued that a stream of moist air came from the south-east through the La Paz valley towards La Paz city. Once the stream reached the end of the valley, it was lifted by thermal and orographic effects. Upon reaching a given height
of about 4000 meters above sea level, moisture began to condensate. These factors, coupled with atmospheric instability, generated a super-cell over the city (Soruco, 2012). This explanation might sound trivial for a super-cell formation but a formal study about the atmospheric characteristics of this precipitation and flash-flood event has never been done.

The goal of this paper is to better understand the atmospheric processes leading to this hail and flash flood episode over the west side of the Andes-Amazon interface. For this purpose, here we will conduct several high resolution numerical simulations.
In section 2 we present the datasets and methods used, section 3 shows the main results including the large and mesoscale conditions observed during the event and results from numerical experiments. We discuss the main findings in section 4 and we summarize the main conclusions in section 5.

## 2 Data and Methods

In order to study the atmospheric characteristics of this hailstorm, we combine satellite and stations observations with numerical
studies. We focus on the time period preceding the cell formation on 19 February 2002 at 1430 LST. The investigation area includes the Andes-Amazon interface connected by several valleys and the Altiplano region near Titicaca lake.

### 2.1 Datasets

#### 2.1.1 Meteorological conditions from Reanalysis

For assessing large scale meteorological features, we use the ECMWF global reanalysis ERA-interim (Dee et al., 2011). It has
a temporal resolution of 6 hours and a spatial resolution of around $0.75° \times 0.75°$ lat-lon with 60 vertical levels. We take the geopotential fields at 200 hPa, and specific humidity and winds at 500 hPa from the 19 February 2002 at 1400 LST file. The ERA-interim dataset is also used as initial and boundary conditions for the following numerical simulations. The simulation horizontal domains are shown in Fig. 1a.





### 2.1.2 Satellite Information

We collect satellite image data from the NOAA Geostationary Operational Environmental Satellite (GOES-8) Imager in the visible channel (0.55-0.75 μm). The spatial resolution is 1 km and Full Earth-Disk images are produced 8 times per day. We take the images at 745, 1115, 1345 and 1745 LST, in order to investigate early shallow convection stages and the hailstorm evolution. We use the Tropical Rainfall Measuring Mission TRMM 3B42 version 7 satellite product (Huffman and Bolvin, 2013) to complement the GOES images. The TRMM Multisatellite Precipitation Analysis (TMPA) are available on a 3-h temporal resolution at $0.25° \times 0.25°$ lat-lon spatial resolution. Despite known uncertainties of precipitation estimates over complex terrain (Rasmussen et al., 2013), they provide area-wise estimates with a fair temporal resolution. This information is particularly useful in remote regions with low weather stations density like the Bolivian Andes. We take the finest simulation domain region (D4 in Fig. 1a is equivalent to all maps shown in Fig. 2) for this purpose.

### 2.1.3 Weather Stations Network

The nature of this event demands a high spatio-temporal resolution precipitation dataset to reveal important characteristics of the hailstorm. The city of La Paz possesses a relatively high spatial (around 1 km distance) but low temporal (24 hours) resolution network of rain gauges; the network is maintained by SENAMHI. We use the available data over the region covered in Fig. 3a. Many of the stations are located on the hill slopes in the city.

Some data quality issues that affects large fractions of station datasets are well known in this region. Nevertheless recent efforts (Hunziker et al., 2017, 2018) have addressed these problems and have produced flags that evaluate the quality of the observations. We use the original dataset taking into accounts the quality flags.

## 2.2 Methods

### 2.2.1 WRF high resolution simulations

The Weather Research and Forecasting Model WRF-ARW version 3.9.1.1 (Skamarock et al., 2008) is used to investigate the physical mechanisms of convection. WRF is a non-hydrostatic next-generation mesoscale numerical weather prediction system designed for both atmospheric research and operational forecasting needs.

The study of a hailstorm atmospheric characteristics requires high resolution simulations. We define four one-way nested domains over the Bolivian central Andes D1, D2, D3 and D4 of 54, 18, 6 and 2 km of grid size, respectively (Fig. 1a). This configuration allows an explicit treatment of deep convective processes in the finest domain. All domains uses Mercator projection and 60 vertical levels with a top level at 50 hPa. The most relevant model configuration can be found in Table 1 and is detailed as follows; we use the Thompson microphysics scheme (Thompson et al., 2008), the YSU planetary boundary layer scheme (Hong et al., 2006) for turbulent fluxes, the Noah land surface model (Ek et al., 2003), and the Rapid Radiative Transfer Model (Mlawer et al., 1997) for long and short-wave radiation. The Kein-Fritsch scheme (Kain, 2004) is used for





cumulus parametrization for domains D1, D2 and D3; being turned-off in D4 for explicit cumulus treatment. The initialisation time is fixed to 1400 LST on 17 February 2002, allowing enough spin-up time until the event.

We couple in addition the one-dimensional hail growth model WRF-HAILCAST (Adams-Selin and Ziegler, 2016) integrated into the WRF-ARW, in order to explore its capabilities for hail production. This approach was used with success in a recent
study in the alpine region (Trefalt et al., 2018) and here we test it in the very high tropical Andes. This main configuration (control run, hereafter CTRL) is used to investigate the physical processes leading to the hailstorm formation and it remain fixed for all set of simulations (sensitivity experiments).

The sensitivity experiments are all initialized at the same time as the control run (1400 LST on 17 February 2002) and they are useful to asses the individual importance of the main features in the region: lake, orography and surface heating. The main
differences with the control run and the goals of each experiment are summarized in Table 2.

We asses the role of orography by modifying the terrain. The Smoothed Terrain Experiment (SMTR) has the valley partially filled up but enlarged, it reduces also (but not drastically) the mountain peaks above the lake. The Reduced Terrain Simulation (RDTR) explores the Altiplano circulation under reduced mountain heights above the lake level following the function,

$$
H_n = \begin{cases} H_r - \frac{H_r - H_L}{2}, & \text{if } H_r \geq H_L. \\ H_r, & \text{otherwise.} \end{cases} \tag{1}
$$

where, $H_n$ is the new orography, the real orography is $H_r$ and the lake level is $H_L$.

We study he role of land surface fluxes for convection development by turning off the energy fluxes between land and atmosphere (NOHEAT experiment). Furthermore, we investigate the lake effects by adding to the lake surface temperature 3 °C (experiment LK+3). The last experiment consists of removing the lake and replacing it by the surrounding land surface type (experiment NOLAKE)

## 2.2.2 Hailstorm diagnostics

We assess the presence of the main ingredients for a hailstorm to occur (moisture, instability and lifting) through a series of atmospheric variables and diagnostics derived from the model output.

The low level moisture transport vectors were calculated following,

$$
IVTU = \frac{1}{g} \int_{SFC}^{200} qu\,dp \tag{2}
$$

$$
IVTV = \frac{1}{g} \int_{SFC}^{200} qv\,dp \tag{3}
$$

with $g = 9.81 \text{ ms}^{-2}$ the gravity constant, $q$ as the specific humidity in $\text{gkg}^{-1}$, $u$ and $v$ as the horizontal wind in $\text{ms}^{-1}$ and $dp$ in hPa as the pressure thickness. It is calculated from the surface $SFC$ up to 200 hPa.





The instability is assessed by the calculation of the convective available potential energy (CAPE). We also asses the surface heat fluxes contribution to buoyancy and explore the lifting mechanisms using vertical speed as a proxy.

## 3   Results

During the hailstorm and flash flood event's precedent weeks, several heavy precipitation episodes were registered over La Paz
city. The continuous water contribution from rainfall kept the soil saturated and limited the absorption capabilities, favouring surface runoff (Hardy, 2009). In the following we present the atmospheric situation of the event itself.

### 3.1   Storm evolution

#### 3.1.1   Synoptic Situation

On 19 February 2002 at 1400 LST, the well known anticyclone at 200 hPa (also called Bolivian High) was located over the
north-east part of Bolivia (Fig. 1b). The intensity and position of the Bolivian High usually drives the large scale moisture transport towards the Altiplano. A rather northern location allows the establishment of a westerly wind circulation which suppresses moist air transport from the Amazon towards the Altiplano (Garreaud et al., 2003).

The atmospheric conditions at 500 hPa confirms the presence of westerly winds over La Paz city in part due to the northward displacement of the Bolivian High and also because of the presence of a strong anticyclone over the Pacific Ocean (not shown).
We find a considerable amount of water vapour over the Bolivian Altiplano due to the continuous precipitation episodes registered during precedent weeks.

This synoptic configuration is favourable for isolating the Altiplano from Amazonian influences, allowing the development of mesoscale features in the presence of a sufficient amount of humidity. It is still unknown how frequently this synoptic configuration occurs during thunderstorms and hail events, since a formal circulation classification is still lacking over the
Central Andes.

#### 3.1.2   Cloud and precipitation evolution from Satellite and Model

Satellite images from GOES-8 describe the fast development of the hailstorm from early morning around 800 LST every 3 hours until late afternoon around 1700 LST. These images are superposed with TRMM rainfall estimates (Fig. 2a-d) and they show a remarkable spatial consistency between each other.
Early morning (Fig. 2a) was characterized by cloudiness over Titicaca Lake, the Amazon region and the eastern cordillera, TRMM is able to capture convective rainfall over the Amazon and near the cordillera; the presence of low level water vapour is not well captured in this band but it's corroborated with infra-red image at 12 μm (not shown). The late morning image (Fig. 2b) reveals the start of important convection over the cordillera and shallow convection over the Altiplano; TRMM is not able to capture any light rainfall and the low level water vapour is still present (not shown). The morning shallow convection then
turned into deep convection during early afternoon (Fig. 2c) with two important cells captured by TRMM at the east of lake



Titicaca and surrounding La Paz city. The northern cell was located southwards from the cordillera and it corresponds very well to the hailstorm location, while the southern cell is located around complex orography inside the Altiplano (called hereafter serranias). At this point the infra-red images are almost the same as the visible channel (not shown). Finally the convective cloud development arrives to its term during late afternoon (Fig. 2d). The Altiplano was then almost completely covered by

clouds and TRMM shows important rainfall localized all over the cloud cover.

As a quick assessment of the model's ability to reproduce the main atmospheric features, we show the model's outgoing long-wave radiation in $Wm^{-2}$ and WRF's 3-hour accumulated rainfall estimations (in $mm$) in Fig. 2e-h. Morning is characterized by high water vapour content and disperse rainfall. We note that the model's rainfall spatial distribution corresponds very well to the clouds locations in Fig. 2a-b) over the Altiplano and cordillera, and less over the Amazon. Early afternoon (Fig.

2g) shows important water vapour at the northern cordillera and a good similarity to Fig. 2c over the Altiplano in term of cloudiness. WRF's late afternoon (Fig. 2h) characteristics are very similar to observations (Fig. 2d)

We conclude that the development of the cell leading to the hailstorm event is mainly due to mesoscale features starting from shallow convection around the Altiplano and later deep convection around complex orography. The cordillera acts as a barrier not allowing much influence from the Amazon, consistently with the synoptic situation. Thus WRF is able to simulate

the event with its most important features.

### 3.1.3 Rainfall estimations from the SENAMHI network

The SENAMHI stations network provides mostly 24h cumulated precipitation measurements from rain gauges. Hourly observations are rare and often incomplete. Nevertheless, they provide an idea of the intensity and the spatial distribution of the rainfall during that particular day. The most important rainfall quantity was therefore registered around La Paz next to the

mountain slopes (with measured values of around 50 $mm$). Some places registered no precipitation, which is a sign of heterogeneous precipitation spatial pattern (Fig. 3a). Consistently with satellite observations, station observations confirms that an important quantity of rainfall fell down close to complex orography and lake.

### 3.2 Physical processes for cell development from Control Run

The analysis of the large scale characteristics and the few observations available provides insufficient information about the

three basic ingredients for a thunderstorm: moisture, instability and lifting. After a confirmation that WRF is able to reproduce the event's main features, we complement the analysis with the output of the CTRL experiment.

### 3.2.1 Topographic features and rain propagation

We therefore explore the model's rainfall propagation over the red lines in (Fig. 3a) in order to explore the chronology of the precipitation. The Hovmoeller diagram along line L1-L2 (Fig. 3b) shows early light rainfall over the north serrania propagating

towards La Paz. The Hovmoeller diagram for line L3-L4 shows a similar behaviour of precipitation but from both sides, the





cordillera and south serrania; forming a short duration rain-band (around one hour) over the Altiplano after propagation (Fig. 3c).

A closer look to the maximum radar reflectivity (in $\mathrm{dBZ}$) spatio-temporal evolution in the model reveals late morning convection in places where lake and/or valley breeze encounter complex orography (Fig. 4a). Later on, the lake breeze becomes

more intense and pushes the rain spots towards the east (Fig. 4b-c); at 1300 LST deep convection is already present and even hailstones of around 5 $\mathrm{mm}$ are simulated at the centre of the two formed cells. The cells finally encounters each other and form a rain-band with isolated hailstorms at the convergence zone between the lake and valley breezes (Fig. 4d-e). Finally the lake breeze becomes weaker and the rain-band dissipates putting an end to this event (Fig. 4f).

### 3.2.2   Low level wind circulation and moisture transport

While the surface specific humidity over the Altiplano follows the lake breeze, the La Paz valley water vapour comes from the Amazon avoiding the cordillera obstacle (Fig. 5a-c). The calculated integrated vapour transport (IVT from surface up to 200 hPa) confirms the partial suppression of moisture transport from the Amazon over the cordillera (Fig. 5a-c). Our results suggest the La Paz valley and Titicaca lake as main humidity sources, with the moisture transport from the lake increasing slightly across time.

Mesoscale conditions a 1100 LST show a rather weak moisture transport following the lake breeze towards the cordillera and both serranias (Fig. 5a). The lake breeze front is accompanied by strong winds at 500 $\mathrm{hPa}$ towards the cordillera and a bit weaker towards the serranias; a strong convergence next to the lake and serrania south also appears (Fig. 5d). Early afternoon (1230 LST) is characterized by stronger IVT from the lake with a small change of direction (Fig. 5b). We observe at the same time an intensification of previous convergence zones around complex orography; with a propagation of the convergence areas

from the previous zones towards each other (Fig. 5e). Around 1400 LST, the lake breeze is dominant (Fig. 5c) and displaces the well formed convergence line towards the edge of the valley (Fig. 5f).

### 3.2.3   Instability

At 1100 LST, instability (indicated by relatively high CAPE values of more than 200 $\mathrm{J\,kg^{-1}}$) develops around the mountain slopes (both cordillera and serrania) and over the Amazon, as shown in Fig. 6a. At the same time, sensible heat is released

in non-cloudy areas (similar spatial distribution as Fig. 4a). During early afternoon at 1230 LST, surface heating intensifies the land sensible energy flux over the Altiplano and La Paz valley, accompanied by an expansion and intensification of CAPE previously identified (Fig. 6b). The highest CAPE values (excluding the Amazon) are located at the north slope of the La Paz valley. Finally at 1400 LST (Fig. 6c) the CAPE intensifies over the valley and sensible heat flux from the surface is released anywhere except the cordillera, serranias and in the proximity if the rain-band.

The city heat-island behaviour is well captured by the model, providing a permanent heat source that contributes to warming the surrounding atmosphere. Skew-T diagrams over La Paz city indicate a stable atmosphere close to saturation at 1100 LST (CAPE of 48 $\mathrm{J\,kg^{-1}}$ in Fig. 6d). The atmosphere then becomes unstable at 1230 LST, with important directional wind shear





and still close to saturation (CAPE of 970 $J\,kg^{-1}$ in Fig. 6e)). At 1400 LST, the atmosphere is saturated until 400 hPa with important wind shear favouring hail and graupel formation (Fig. 6f).

### 3.2.4 Lifting and propagation mechanisms

Previous sections reveal convection initiation around humid and unstable zones near complex orography, resulting in the

formation of two important cells (also detected by satellite). Both cells then propagates towards each other forming a strong convection band. The location of this band overlaps the lake-valley breeze convergence zone. The evolution of the intensity of vertical velocity at 4000 meters above ground level (magl) and wind shear from surface to 6000 magl (Fig. 7a-c) gives an idea about the severity of afternoon convection and resulting storm. The responsible lifting mechanisms identified until now are orography and lake-valley breeze convergence. Therefore, we focus on these areas and explore the vertical structure between

lake and valley (line A-B in Fig. 7a) and between cordillera and serrania (line C-D in Fig. 7a).

The lake-valley cross section confirms the early light convergence over the Altiplano and surface heating (Fig. 7d-f). Nevertheless, convergence is not enough to explain deep convection. The later appearance of a cold pool (Fig. 7f) over the convergence zone (which coincides with the city's location), combined with orographic forcing, may have triggered the deep convection. Low level moisture is then rapidly lifted and encounters a supercooled atmospheric environment, highlighted by

the low freezing level of around 1000 magl (also in Fig. 7f).

The cross section between complex orography shows important surface heating next to mountain slopes during early afternoon (Fig. 7i). Important convection is then initiated over the serrania and propagates towards the cordillera over the lake-valley convergence zone (Fig. 7h-i). Figure 7i reveals later appearances of cold pools over the mountains slopes which, combined with breeze convergence, may explain the multiple zones of deep convection. This cross section shows that the cold pool seen in

Fig. 7f is part of the cold pool that propagated from the cordillera.

These results suggest that the two orographic cells produced rain that cooled down the surface and originated cold pools. The propagation of these cold pools, combined with instability, convergence and orographic forcing, then triggered deep convection resulting in a rain-band that included isolated hailstorms.

### 3.3 Sensitivity studies

After we described the hailstorm dynamics from the control run, we assess the principal elements participating to the storm formation and propagation.

### 3.3.1 Orography influence

The smoothed terrain experiment (SMTR) provides a larger La Paz Valley extent, allowing a better organization and expansion of valley breeze. Since the mountains summits are still high, the orographic convection is still present. Nevertheless, the

stronger valley breeze enhances convergence with lake front breeze and the cold pool lifting is no longer necessary for deep





convection (Fig. 8a-c). The resulting rain region is less organized and the band expands heterogeneously with hail originated by valley breeze and orographic interactions (Fig. 10b)

In the case of the reduced terrain (RDTR experiment), we purposely reduced the mountain heights and observed a drastic reduction of deep convection regions (Fig. 8d-f). Rainfall still exists but it is weaker (Fig. 10c), highlighting the importance of high mountains for convective initiation. Nevertheless, hail can still be produced by the model.

### 3.3.2 Soil Fluxes

In the experiment with the surface heat fluxes suppressed (NOHEAT), convection is still present without thermodynamic instability; but cells are very isolated. The precipitation regions (Fig. 10d) corresponds to complex orography, showing that breeze and orographic lifting are enough for producing rainfall.

### 3.3.3 Lake Breeze

A warmer lake (experiment LK+3) reduces the temperature gradient with the land surface and weakens the lake breeze. Cross section over A-B line shows that the La Paz valley wind becomes predominant and vertical motion is less intense (Fig. 9a-c). This shifts the rainband towards the lake and suppresses hail formation (Fig. 10e).

When the lake is removed (NOLAKE experiment) the surface where the lake was located becomes a shallow valley and the wind circulation is similar to a lake breeze. Nevertheless, the main mechanism for convection is surface heating and orographic lifting (Fig. 9a-c). In this experiment, convection occurs later than experiments containing the lake and it concentrates around the serranias, leaving the cordillera storm free with isolated hailstorms (Fig. 10f).

## 4  Discussion

The iconic heavy hailstorm and flash flood on 19 February 2002 over La Paz city was an exceptional natural disaster that still remains in people's memory. There were several factors that increased the city damage: the hail location, the water management system and the fast cell development.

Satellite images shows early shallow convection over the Altiplano during late morning that became deep convection over complex orography in the early afternoon; forming notably two cells, one over the cordillera and another over the south serrania. The two cells then propagated towards each other, producing strong precipitation and isolated hailstorms. Synoptic conditions were favourable to the mesoscale circulation development at near surface levels, produced mainly by lake breeze and orographic thermal circulation.

Using WRF simulations, we present new insights on the local atmospheric conditions leading to the formation of deep convection and later hailstorm and flash flood episode over La Paz. We focus on the dynamical mechanisms before and during the event under the described synoptic conditions; notably moisture sources, atmospheric instability and lifting mechanisms. The results from CTRL experiment are consistent with satellite and stations observations, synoptic circulation and historical



evidence. The simulations output are able to reproduce the spatial pattern of cloud cover and precipitation. They are therefore able to reveal plausible key processes.

On 19 February 2002, surface wind over the altiplano was guided by thermal lake, mountain and valley breeze effects. The significant precipitation falling on previous days over the region saturated the soil and provided a local moisture source (Hardy, 2011). On top of these already moist conditions, even more water vapour entered the region from the valley and the lake following the thermo-topographic circulation.

The first convective cells were formed during late morning over convergence zones that were located over complex orography. They were triggered by a mix of low level wind convergence, surface heating and orographic forcing. Later on, the atmospheric unstable zones grew in surface and strength with a bigger amplitude over the valley. The surface heating increased the lake-land temperature gradient and exacerbated the lake breeze; keeping the northern cell over the cordillera and pushing it towards La Paz. The southern cell at the same time also grew but stayed stationary.

However, the most remarkable results from WRF simulations concerns the secondary trigger mechanism. It is revealed that the first convective cells formed cold pools that propagated following the mesoscale circulation. This propagation allowed both cells to join each other resulting in a precipitation band. This auto-propagation mechanism has been observed by previous works over the Alps (Trefalt et al., 2018; Kunz et al., 2018) or in idealized situations (Schlemmer and Hohenegger, 2014), but not yet over the central Altiplano.

The presence of sufficient wind shear extends and supports the organization of convective storms in terms of multicells, supercells or mesoscale convective systems. Wind shear is often measured between surface and 6 km above surface and for hail formation they can be moderate (Trefalt et al. (2018) found values of around $10 \, \mathrm{m \, s^{-1}}$), or unusually high (Kunz et al. (2018) observed values of more than $40 \, \mathrm{m \, s^{-1}}$. In our case we find a rather moderate 0-6 km wind shear (around $10 \, \mathrm{m \, s^{-1}}$), but it becomes strong if we take a 0-9 km basis (around $20 \, \mathrm{m \, s^{-1}}$), as shown in Fig. 7f.

The sensitivity experiments highlights the importance of orography for low level wind circulation and for triggering deep convection. A reduced mountain altitude suppresses deep convection while a smoothed but still high orography modifies the upslope and valley wind circulation. They also reveal the crucial influence of surface energy fluxes for atmospheric instability.

The lake breeze effect turns out to be determinant for storm propagation. The sensitivity experiments shows that a higher lake surface temperature results in a smaller temperature gradient between land and lake, which weakens the lake breeze and shifts the convergence zones eastwards. Similar numerical studies found a comparable behaviour over the Tibetan plateau (Gerken et al., 2014). Gerken et al. (2014) found that the lake breeze strength controls the location of convection over a mountain. While we get similar results, we also observe a competition between lake and valley breezes over the convergence line. A lake suppression eliminates the deep convection over the cordillera and highlights the lake breeze role in the northern cell formation.

While hailstorms are perceived to increase in frequency over the Altiplano (Boillat and Berkes, 2013; Yasukawa, 2011); hail hotspots have not been formally identified yet. Nevertheless, several studies have studied the formation of severe thunderstorms in the southern Andes that can be initiated by gravity waves (de la Torre et al., 2015, 2011) or the passage of a cold front (Teitelbaum and D'Andrea, 2015). The common ground with our study is the importance of orography and a supplementary



lifting mechanism. We find two lifting mechanisms in chronological order: early convection is initiated by a mix of breeze and orography, and later deep convection is originated by a combination of cold pool and orographic lifting at the end of valley.

## 5   Summary and Conclusions

This study analyses the main atmospheric characteristics of the historical hailstorm of 19 February 2002 over La Paz City. A

first assessment from scarce station observations, satellite information and reanalysis suggests that this severe event was in fact part of a mesoscale convective system. The large scale moisture transport towards the Altiplano was in part blocked by the high Cordillera at the interface Andes-Amazon due to the position of the Bolivian high. This synoptic condition allowed the formation of mesoscale thermal circulation.

The control run (previously evaluated with TRMM dataset) offers a better insight of the mesoscale atmospheric dynamics

preceding the hailstorm. A first convection episode is generated by thermal instability and triggered by lake breeze and orographic forcing. The resultant precipitation then forms two cold pool clusters over the mountain slopes. The Cordillera cold pool is shifted towards the east by the lake breeze and middle level westerly circulation and results in a convergence rain band from lake and valley breeze. The La Paz cold pool stay stationary and keeps cooling, lowering the freezing level and triggering a secondary, deeper convection to the north of La Paz.

Additional sensitivity experiments show an enhanced valley breeze effect from the north in presence of warmer lake surface, which also provides more moisture through evaporation, resulting in an earlier stronger rainfall (LAKE+3). A lake suppression (NOLAKE) modifies the thermal circulation and eliminates deep convection over the cordillera. The smoothed terrain experiment (SMTR) shows the importance of the altitude of the Andes-Amazon interface mountain chain in the moisture transport blocking and on the strength of the valley breeze. The reduced terrain simulation (RDTR) shows the importance of high moun-

tains for convection initiation. And the surface heat flux suppression (NOHEAT) highlights the importance of surface energy fluxes for atmospheric instability.

The analysis conducted in this study highlights the complex interaction between large scale circulation, orography and local features in the formation of hailstorms over the tropical Altiplano. A semi-comprehensive scheme of participating mechanisms can be found in Fig. 11.

Our results are consistent at the same time with the few low resolution observations, the historical context and the known atmospheric dynamics; showing that it is possible to complement the observations with numerical techniques in order to better understand local processes resulting in severe weather over the tropical Andes' complex orography.

The combination of large scale analysis and high resolution numerical techniques used in this study offer new elements for forecasting purposes. Nevertheless, we stress that the proposed mechanisms of this hailstorm formation should be confirmed

by high resolution observations and further numerical investigations of similar high-impact events.



*Acknowledgements.* This work received the support from the Federal Commission for Scholarships for Foreign Students through the Swiss Government Excellence Scholarship (ESKAS No. 2015.0793) for the academic year(s) 2015-18/19. It was also supported by the project "Servicios CLIMáticos con énfasis en los ANdes en apoyo a las DEcisioneS" (CLIMANDES), no. 7F-08453.01, funded by the Swiss Agency for Development and Cooperation (SDC) and coordinated by the World Meteorological Organization (WMO).





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




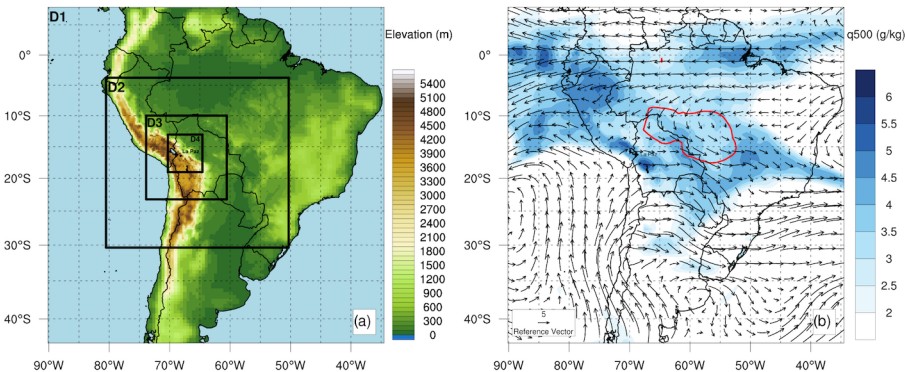

**Figure 1.** (a) Experiment spatial configuration with orography as seen by the model. (b) 19 February 2002 at 1400 LST synoptic conditions from ERA interim. Red contour indicates the position of the Bolivian High at 200hPa; 500 hPa conditions are highlighted by wind circulation [m s$^{-1}$] (curly vectors) and specific humidity [g kg$^{-1}$] (blue filled contour)

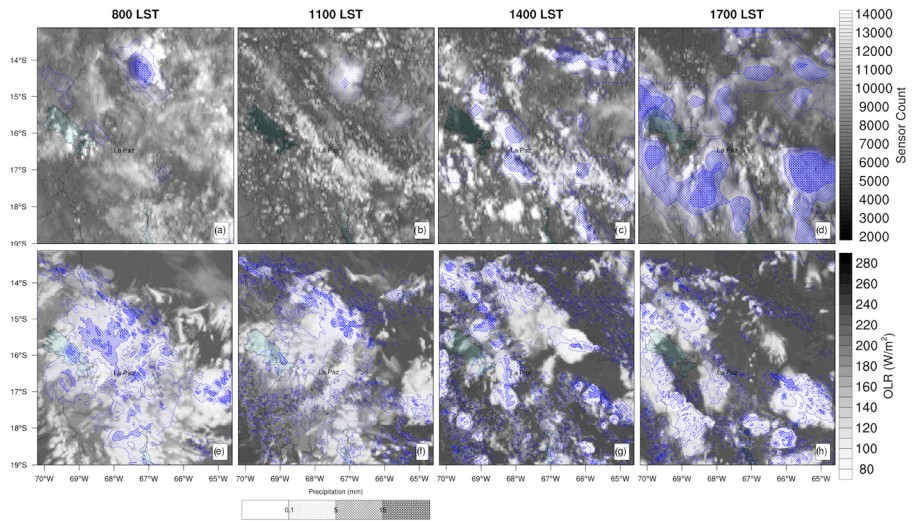

**Figure 2.** (a)-(d) Remote Sensing observations assessment: white filled contour shows GOES-8 observations in visible band [counts] and blue shaded contour is TRMM 3-hour accumulated rainfall estimations in mm. (e)-(h) Model outgoing longwave radiation [W m$^{-2}$] in white filled contour and 3-hour accumulated rainfall estimations [mm] in blue shaded contour.





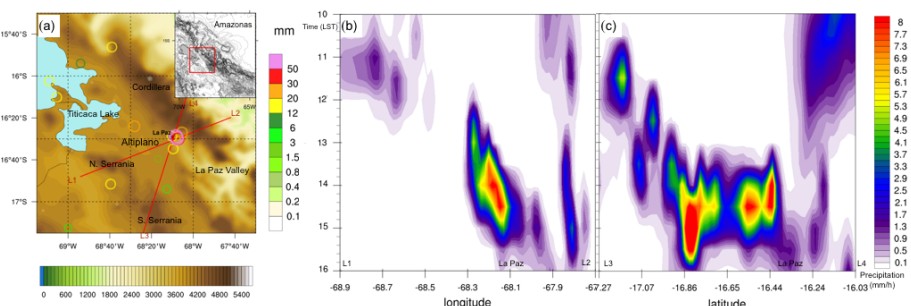

**Figure 3.** (a) Topographical features and 24h cumulated rainfall distribution from SENAMHI network in mm. Model rainfall Hovmoeller diagram [mm h$^{-1}$] propagation across the lines (b) L1-L2 and (c) L3-L4 .

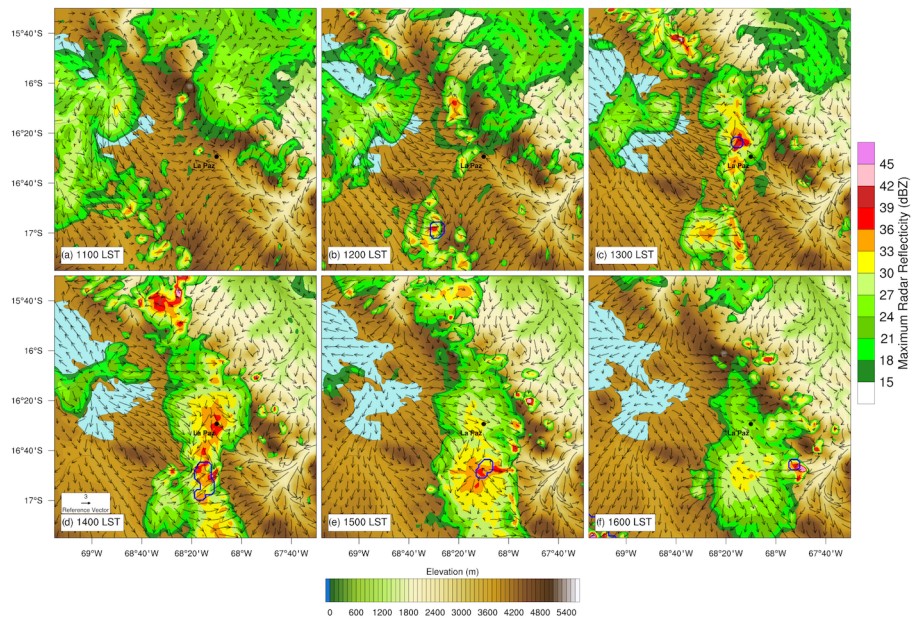

**Figure 4.** Hourly evolution of maximum radar reflectivity in dBZ (coloured filled contour), hailstone with a diameter of around 5 mm size regions (blue contour) and 10 meter wind speed in m s$^{-1}$ (curly vectors).





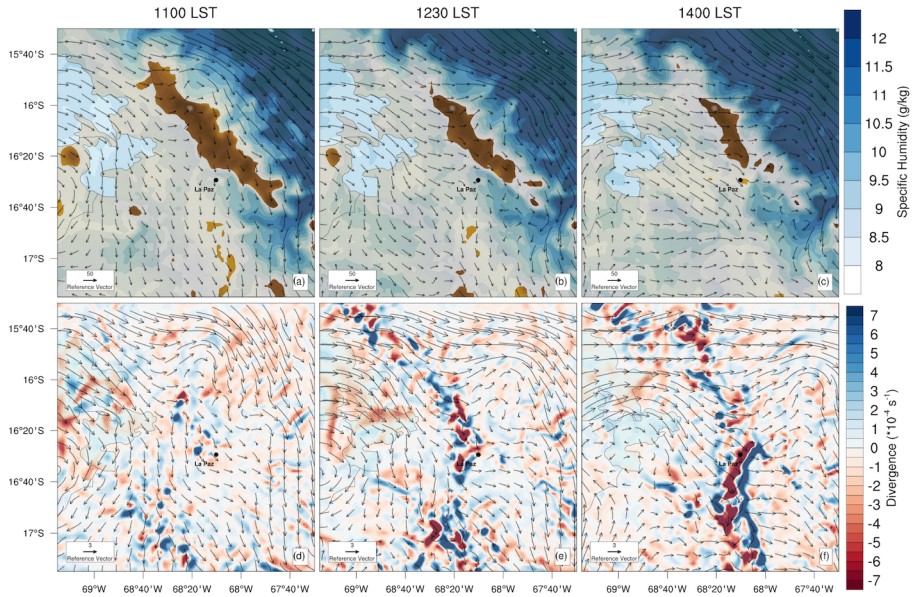

**Figure 5.** (a)-(c) IVT from surface to 200 hPa [kg m$^{-1}$s$^{-1}$] (curly vectors) and specific humidity [g kg$^{-1}$] at 2m (filled contours). (d)-(f) Wind speed [m s$^{-1}$] (curly vectors) and divergence [$10^{-4}$s$^{-1}$] (filled contours) at 500 hPa.

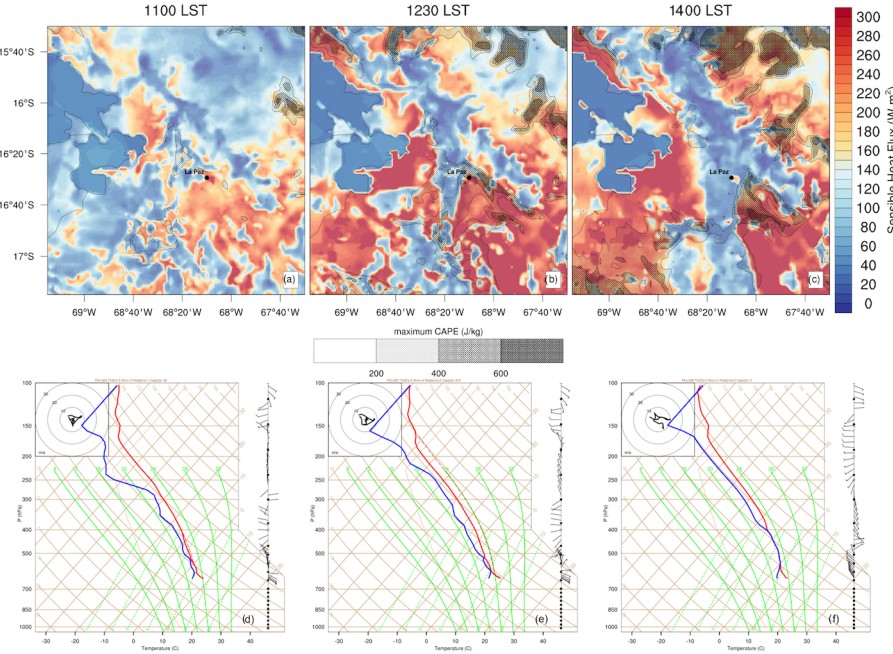

**Figure 6.** (a)-(c) Surface sensible heat flux [W m$^{-2}$] (filled contours) and maximum CAPE [J kg$^{-1}$] (shaded contour). (d)-(f) Simulated Skew-T diagrams and hodographs over La Paz city.



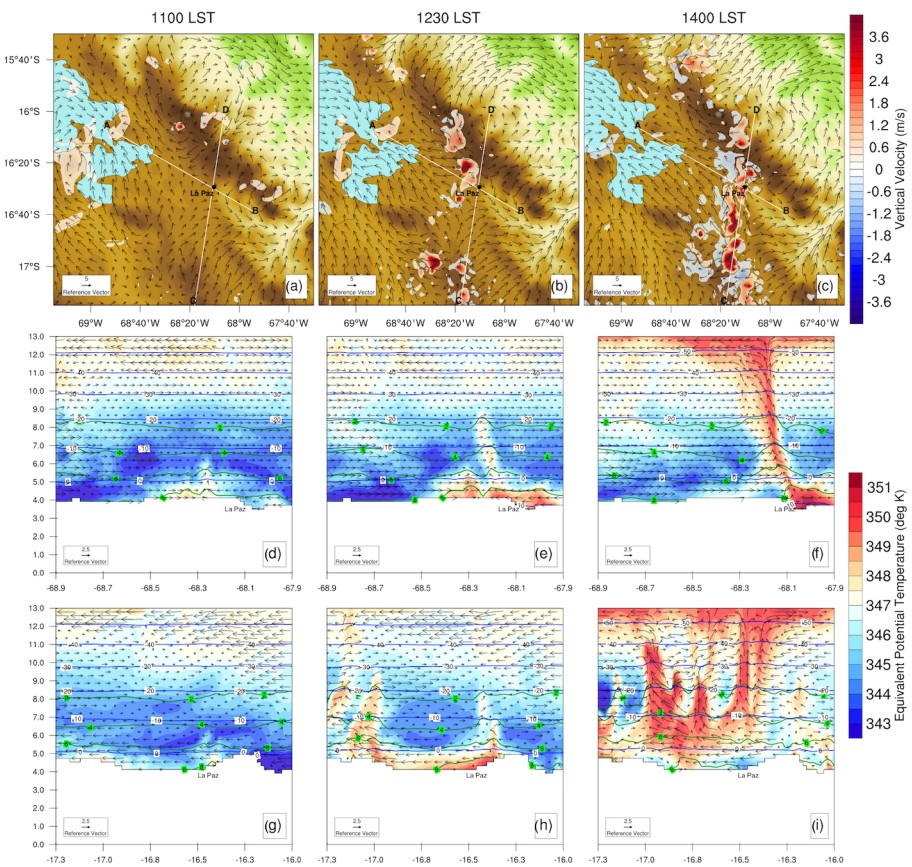

**Figure 7.** (a)-(c) Vertical velocity [m s$^{-1}$] at 4 km above ground level (agl) (filled contours) and wind shear from [m s$^{-1}$] surface until 6 km agl (curly vectors). (d)-(f) Vertical cross section along A-B white line, including $\theta_e$ [deg K] (coloured filled contours), vapour mixing ratio [g kg$^{-1}$] (green contour), air temperature [deg C] (blue contour) and wind velocity [m s$^{-1}$] (curly vectors). (g)-(i) is the same as (d)-(f) but across C-D white line.




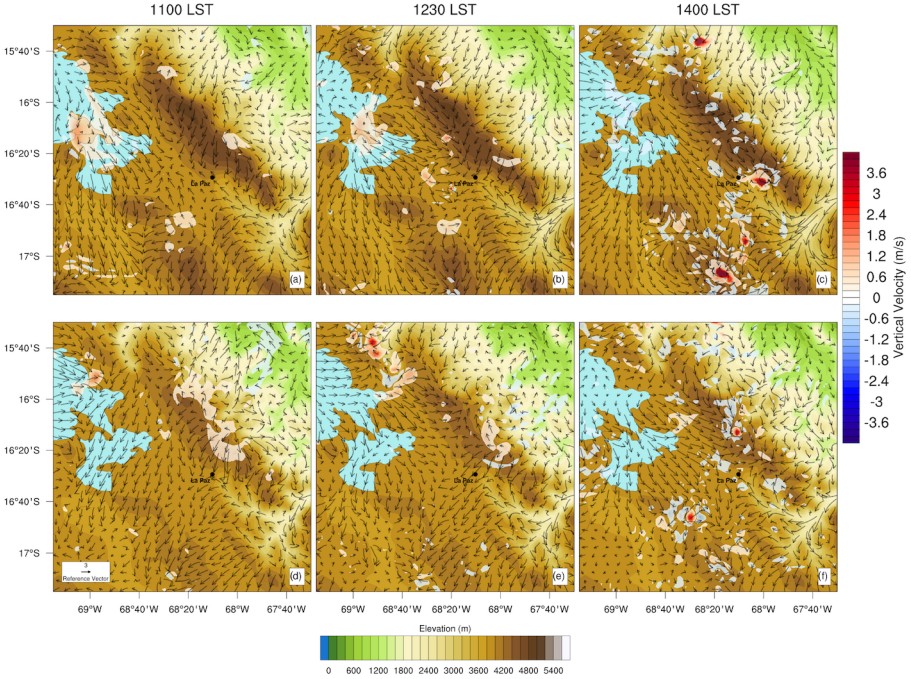

**Figure 8.** Vertical velocity [m s$^{-1}$] at 4 km above ground level (agl) (filled contours) and 10 meter wind speed [m s$^{-1}$] (curly vectors) for experiment SMTR ((a)-(c)) and RDTR ((d)-(f))

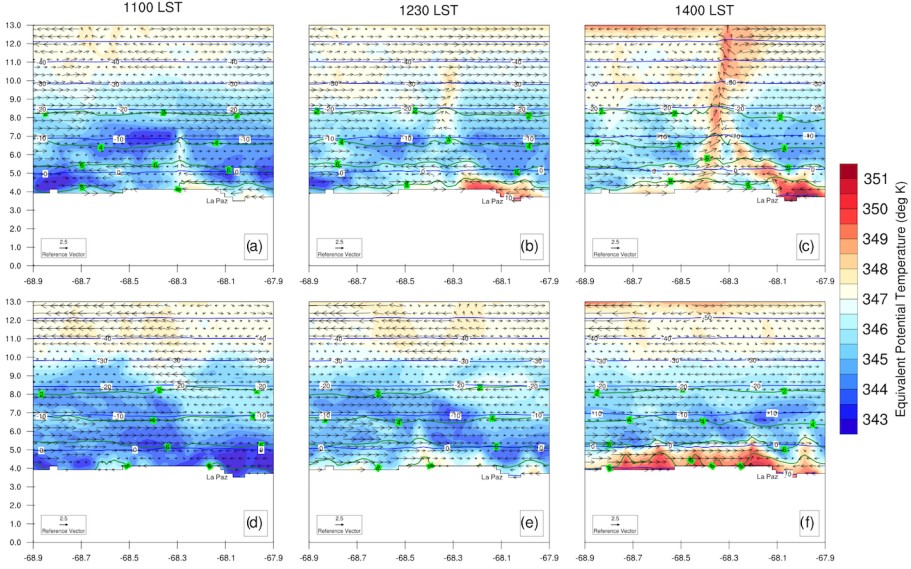

**Figure 9.** As for Figure 7d)-f) but for experiments LAKE+3 ((a)-(c)) and NOLAKE ((d)-(f))




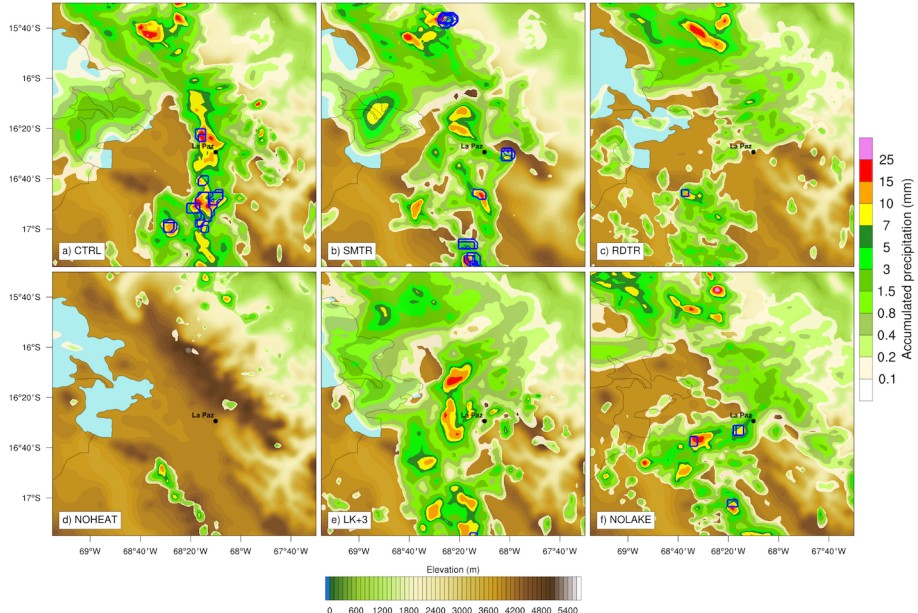

**Figure 10.** Model accumulated rainfall in mm from 1200 to 1500 LST for runs (a) CTRL, (b) SMTR, (c) RDTR, (d) NOHEAT, (e) LK+3 and (f) NOLAKE. See Table 2 for a description of the simulations (filled contour corresponding to vertical labelbar). Model terrain is included (filled contour corresponding to horizontal labelbar). Blue contours indicate the locations of simulated hailstones with diameter between 4 and 6 [mm] at the same temporal range.

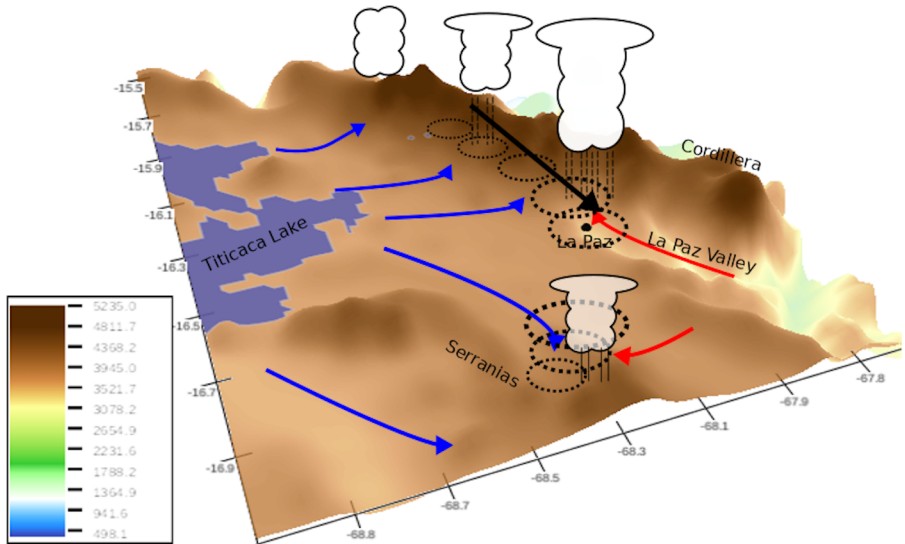

**Figure 11.** Schematic diagram of principal factors contributing to hailstorm development. Brown shading represents the orography, blue arrows the lake breeze, red arrows the valley breeze, dotted circles the cold pool locations, and black arrows the cold pools propagation. Clouds and rain are also schematically represented.





**Table 1.** Main configuration used in WRF simulations

| Parametrization | Scheme |
|---|---|
| Cloud microphysics | New Thompson scheme |
| Planetary boundary layer | Yonsei University scheme |
| Land surface model | Noah land surface model |
| Radiation | Rapid radiative transfer model for longwave and shortwave |
| Cumulus parametrization | Kein-Fritsch scheme, parametrization off for D4 |

**Table 2.** Summary of WRF experiments

| Name | Goals | Description |
|---|---|---|
| CTRL | Real data configuration | Real orography, land use, and lake temperature |
| SMTR | Sensitivity to terrain smoothing | Terrain smoothed by 7 times 1-2-1 filtering |
| RDTR | Sensitivity to terrain reduction | Terrain above lake level reduced by half |
| NOHEAT | Sensitivity to land heating | Sensible heat fluxes between land and atmosphere turned off |
| LK+3 | Sensitivity to lake temperature | Modification of lake surface temperature by +3 [deg C] |
| NOLAKE | Sensitivity to lake presence | Replacement of lake by the surrounding land use |