# Peer review of "Characteristics of a Hailstorm over the Andean La Paz Valley"

_Natural Hazards and Earth System Sciences, 2019_

## Referee Comment (RC1) · Anonymous Referee #1 · 8 Mar 2019

**NATURAL HAZARDS AND EARTH SYSTEM SCIENCES**
**Manuscript NHESS-2019-27**

**CHARACTERISTICS OF A HAILSTORM OVER THE ANDEAN LA PAZ VALLEY**

**OVERVIEW:**
This investigation analyzes the occurrence of a major hailstorm accompanied also by flash flood that occurred over La Paz and vicinities in February 2002. Despite the scarcity in available observations, the authors make an effort to describe the atmospheric conditions that led to the development of the severe weather event. To that end, TRMM and GOES satellite imagery, measurements from rain gauges, and gridded output from ERA Interim Reanalysis and from high resolution simulations with ARW-WRF model were employed. In addition, a sensitivity analysis based on a set of simulations with ARW-WRF was performed to assess the role played by the local geography (through terrain elevation and land-atmosphere interaction) in influencing the atmospheric environment in which the severe storm formed. The goals of the investigation are relevant, especially when one considers the lack of studies addressing severe local storms in Bolivia. However, a number of important issues must be addressed before the manuscript can be considered ready for publication. I decided to rate the manuscript as **reconsidered after major revisions**. Below the authors will find my detailed comments.

**TITLE:**
The title makes reference only to the hailstorm, but throughout the text the authors also mention/discuss the occurrence of a flash flood accompanying the same weather event. Therefore, the title should be modified, perhaps by reading "Characteristics of a Severe Convective Storm over...". The main point is that the authors give equal emphasis to the hail precipitation and to the flash flood (rainfall amount) in the text, but the title, as it is, does not reflect that.

**FIGURES:**
Most figures are appropriate for describing the results, but they are way too small, making it hard to read and to verify several of the important detailed information discussed in the text. It is true that the digital file allows for the zooming in of the figures, but this is rather cumbersome for the reviewer. If, for example, the authors keep one figure per page (in the submitted version) then the figures can be enlarged.

Captions can be improved and/or do not provide full information of the contents of the figures:
Caption of Fig.1a must inform the horizontal grid spacing for each domain.
Caption of Fig.2 begins with *"(a)-(d) Remote Sensing observations assessment"* which could be replaced by "(a)-(d) GOES-8 visible imagery (grey shading) and TRMM estimated 3-hour accumulated precipitation (blue shading)..."
Caption of Fig.3: "accumulated" instead of "cumulated".
Caption of Fig.4: Should read: "...maximum simulated radar reflectivity in domain D4..."
Caption of Fig.4: Should read: "...blue contour encloses areas with simulated hailstones equal to or larger than 5 mm in diameter...", and must inform at what vertical level this is valid. Surface level?
Caption of Fig.6: Should read: "...most unstable CAPE..." instead of "...maximum CAPE..."

**INTRODUCTION:**
**Page 1, line 24:**
*"...between 1420 and 1545 LST [...] a hailstorm affected the city of La Paz."*
Please, inform the corresponding UTC times as well. Has the hail precipitation lasted for 1 hour and 25 minutes over La Paz? That would be highly unusual; a trully extreme event. Or was the accompanying flash flood that lasted for such a long period?

**Page 2, line 11:**
*"...generated a super-cell over the city (Soruco, 2012). This explanation might sound trivial for a super-cell formation..."*
As for a supercell being reponsible for the hailstorm, it surprises me that the authors of this study run a fairly high resolution simulation of the convective storms with WRF but do not verify whether any of the simulated cells developed a mesocyclone. That could provide additional evidences for the supercellular nature of the storm(s). The authors should look for such evidences in the 2 km grid-spacing simulations through the analysis of convective updrafts correlated with (negative) vertical vorticity. More detailed comments on that matter follow below.

**DATA AND METHODS:**
**Page 2, line 25:**
"*...a temporal resolution of 6 hours and a spatial resolution of around 0.75 x 0.75 lat-lon...*"
I am not sure that we can state that the temporal resolution of the ERA Interim is of 6 hours since we would need at least 2 "time-steps" (i.e., 12 hours in this case) to minimally resolve any atmospheric feature using this dataset. The same comment holds for the spatial "resolution".
I suggest rephrasing by "...the gridded data is available at 6-hour intervals..." and by "...with horizontal grid spacing of 0.75° x 0.75° latitude-longitude...".

**Page 2, line 26:**
"*...geopotential fields at 200 hPa, and specific humidity and winds at 500 hPa...*"
The authors extract 200hPa geopotential fields from ERA Interim but never show these fields explicitly.

It must be indicated what is the above-ground height of the 500 hPa pressure level over La Paz. This is important because, at first, it sounds strange to analyze the 500 hPa humidity fields when we should be mostly interested in the analysis of the low-level moisture (below 3000 m AGL). It turns out, however, that La Paz is situated in very high terrain and therefore the 500 hPa fields may represent the (local) low-troposhere, which is unusal for most regions.

**Page 3, line 8:**
"*...they provide area-wise estimates with a fair temporal resolution...*"
I would rather state more explicitly that the 3-hr sampling interval from the TRMM satellite, despite **not** being adequate for monitoring the evolution of a single severe convective storm, is the best available remote sensing data for this specific case study.

The authors only utilized the rainfall estimation product from TRMM satellite. Given the severity of the storm, other products could have been analyzed, such as the height of the 40dBZ radar reflectivity just as one example. South American hailstorms are known for being very tall, particularly in the La Plata Basin sector. Most readers will be curious about the depth of this cell in Bolivia; has TRMM sampled the storm at its mature stage?

**Page 3, line 14:**
"*...resolution network of rain gauges; the network is maintained by SENAMHI.*"
Is this an automated surface network? Or is it manned? This must be informed for the sake of completeness.

**Page 3, line 25:**
"*...over the Bolivian central Andes D1, D2, D3 and D4 of 54, 18, 6 and 2 km of grid size...*"
I wonder if the D1 domain with 54 km horizontal grid spacing is really necessary when downscalling from ERA Interim. The downscale "leap" from ERA Interim directly to the 18 km grid spacing may had sufficed. Any comments on that?
Please, provide the number of gridpoints (matrix size) of the 2 km mesh.

**Page 3, line 30:**
Mispelling:
"The KAIN-Fritsch scheme...."

**Page 4, lines 1-2:**
"*The initialisation time is fixed to 1400 LST on 17 February 2002, allowing enough spin-up time until the event.*"
First question: 14:00LST = 18:00UTC?

I understand the authors´ concern with the model´s spin-up period but, in my experience and from several other numerical studies on convective storms, initializing the simulations 24-hr before the convective event usually suffices for that matter. Starting 48-hr in advance (as done here) may lead to too long a "forecast range" to produce the best possible simulation. Have the authors tested distinct initialization times for the simulations? If so, was the choice of utilizing the one starting 48-hr before the event justified for being the simulation with best correspondence with observations?
Finally, were all four domains initialized at the same time? These pieces of information should be informed.

**Page 4, line 20:**
The authors have available the output of a fairly high resolution WRF simulation (their domain D4) of the convective storms, but as "hailstorm diagnostics" they follow an ingredients-based approach ("*We assess the presence of the main ingredients for a hailstorm to occur...*") for which having a high-resolution simulation is not indispensable. I recognize the importance of the ingredients-based approach, but additional diagnostics should have been chosen that explore the full explicit information made available by the high resolution simulations. Interestingly, in the Results section, the authors do show variables/fields such as simulated reflectivities, updraft strength, surface winds, and areas enclosed by hailstones surpassing a given diameter threshold, but none of these variables/fields is mentioned in the methodology as a diagnostic.

The parameter "updraft helicity", computed around 3 km A.G.L., would be also a natural choice of diagnostic to verify if the simulated storm(s) displayed mesocyclones (i.e., if they behaved as supercells) in any given stage of its(their) development. At least, vertical velocities should be analyzed in tandem with vertical vorticity in order to assess the presence (or the lack thereof) of mesocyclones. Surface winds/outflow produced by the simulated storms are shown in the Results section but could be better utilized by the authors when assessing the storms´ severity.

Finally, the presence of moderate to strong vertical wind shear is among the typical ingredients for severe convective storms, but the authors do not include any parameter for vertical wind shear in this section, despite discussing this parameter in the Results section.

**RESULTS:**
**Page 5, lines 9-10:**
"*...the well known anticyclone at 200 hPa (also called Bolivian High) was located over the north-east part of Bolivia (Fig. 1b).*"
How the Bolivian High was characterized? The authors do not show the 200 hPa geopotential heights in Fig.1b.
To a large extent, the Bolivian High is a response to the intense convective activity (latent heating) observed over central South America during the warm season, so it is as much a consequence from deep convection than the cause for it. The discussion in Section 3.1.1 indicates that the Bolivian High drives/influences the convective activity but does not stress the important feedback from the convection itself.

**Page 5, lines 15-16:**
"*We find a considerable amount of water vapour over the Bolivian Altiplano due to the continuous precipitation episodes registered during precedent weeks.*"
Shouldn´t the presence of water vapour over the Bolivian Altiplano be the cause for the precipitation events rather than a consequence from it?

If the Amazon Basin was not the moisture source for the Bolivian Altiplano (as stated by the authors in lines 11-12 of page 5), what was the effective moisture source? I know the authors discuss this matter in more details later on in the text, but my point here is that the general perspective provided by Fig.1b alone does not convince the reader that the Amazon Basin was not a moisture source for the Bolivian Altiplano.

Fig.1b also suggests the presence of the South Atlantic Convergence Zone; do the 850hPa fields (not shown) also depict that?

**Page 5, line 22:**
"*Satellite images from GOES-8 describe the fast development of the hailstorm...*"
Figs.2a-d *per se* do not allow the identification of the hailstorm. Maybe an arrow could be superimposed to the image to indicate which cell is the hailstorm; or else the figure caption should inform that.

**Page 5, lines 26-27:**
"*...the presence of low level water vapour is not well captured in this band but it's corroborated with infra-red image at 12 μm (not shown).*"
I do not agree with this specific statement. The thermal infrared imagery at 12 μm is useful to detect clouds and storms with tops at distinct heights, but not to detect low-level water vapour. In fact, it is hard to detect low-level water vapour from the geostationary satellite imagery, with the most reasonable choice (with GOES 8) being at mid-levels utilizing the 6.48 μm channel ("water vapour channel").

As for the 12 μm channel, was the hailstorm exceptionally deep for the Altiplano region (as inferred from the brightness temperature)?

**Page 5, line 30 and page 6, line 1:**
"*...with two important cells captured by TRMM at the east of lake Titicaca and surrounding La Paz city.*"
Again, it is hard to identify these cells in Fig.2c. The authors should try to superimpose arrows to the satellite imagery to highlight the convective cells being of most interest.

**Page 6, lines 1-2:**
Here the authors mix two verb tenses "northern cell was" and "southern cell is". Please, choose one verb tense when describing the event and stick to it throughout the text.

**Page 6, line 3:**
"*At this point the infra-red images are almost the same as the visible channel (not shown).*"
I cannot understand what the authors mean by this statement. It is best to remove it since it is confusing and does not add relevant information.

**Page 6, lines 3-4:**
"*...the convective cloud development arrives to its term during late afternoon (Fig. 2d).*"
I would suggest rephrasing to "*...the demise of the convective activity occurred during the late afternoon (Fig.2d).*"

**Page 6, lines 7-8:**
"*Morning is characterized by high water vapour content and disperse rainfall.*"
The simulated radiance from WRF does not inform "water vapour content", but provides a simulated image from the thermal infrared band which is utilized to detect brightness temperatures from distinct surfaces and cloud tops, implying (in the case of clouds) the presence of hydrometeors and not simply water vapour.

**Page 6, lines 8-9:**
"*...the model's rainfall spatial distribution corresponds very well to the clouds locations in Fig. 2a-b over the Altiplano and cordillera, and less over the Amazon.*"
It seems clear to me that the simulation overestimated the cloud cover/rainfall to the east of Lake Titicaca and over the Altiplano and Serranias. Moreover, the strongest simulated cell at 0800LST (Fig.2e) was located south-southwest of the respective observed cell (Fig.2a). I generally do not expect the model to nail down the exact location and timing of the convective storms, but I do not agree with the statement that "*the model´s rainfall spatial distribution corresponds very well to the clouds locations*"; in fact, the misplacement of the strongest cell at the early stages of the weather episode may explain some of the surface features displayed in the following figures and should be stressed in the text.

**Page 6, lines 9-10:**
"*Early afternoon (Fig. 2g) shows important water vapour at the northern cordillera...*"
Again, Fig.2g does not show water vapour. If the authors wish to describe the behaviour of the atmospheric water vapour in the simulation then they must plot the simulated water vapour mixing ratio (or specific humidity), not the simulated outgoing long wave radiation.

**Page 6, lines 12-15:**
In this paragraph the authors jump into two conclusions without presenting solid arguments to back them up. First, that the hailstorm was mainly induced by mesosale features, and, second, that the cordillera blocked the moisture flow from the Amazon. At this point they can only hypothesize these two aspects. The authors should fisrt describe the WRF simulations in more details before presenting these conclusions.

**Page 6, Section 3.1.3:**
I think the discussion in this Section is poor. First, there is no figure illustrating the analysis; second, the authors should provide more specific/detailed information instead of just stating "Some places registered no precipitation..." and "...station observations confirms that an important quantity of rainfall fell down close to complex orography...". Around La Paz there is more than one important orographic feature, so what exactly "complex orography" are we talking about?

How did the WRF simulated rainfall compare to the observed rainfall from the rain gauges? How did the TRMM-estimated rainfall compare with the rain gauges? These are relevant information since TRMM was also used to evaluate the WRF simulations in the previous section.

In line 17 it should read "accumulated precipitation".

**Page 6, lines 24-25:**
"*The analysis of the large scale characteristics and the few observations available provides insufficient information about the three basic ingredients for a thunderstorm: moisture, instability and lifting*."
Actually, the limitation goes beyond the ingredients-based analysis: the pieces of information analyzed thus far could not provide any insight regarding the internal strucuture of the convective storms, especially in the absence of a local weather radar. This is a particularly relevant issue considering the study of a severe hailstorm. Therefore, a (good) high resolution numerical simulation is desirable to provide such an important insight.

**Page 6, line 28:**
Avoid starting this paragraph with "We therefore...."

**Page 7, line 3:**
"A closer look AT the maximum SIMULATED radar reflectivity..."

**Page 7, lines 3-6:**
"...*reveals late morning convection in places where lake and/or valley breeze encounter complex orography (Fig. 4a).*"
I do not think the simulated radar reflectivity alone provides this information. Perhaps the simulated surface winds combined with the simulated reflectivity can indicate that, but, still, it is hard to reach a conclusion from Fig.4 alone. To better characterize mesoscale fronts (such as breezes) it would be better to plot the divergence fields at the low levels and look for linearly-oriented convergence features.

"*Later on, the lake breeze becomes more intense and pushes the rain spots towards 5 the east (Fig. 4b-c).*"
Given the highly divergent pattern of the simulated winds over the Titicaca Lake associated with earlier convection (Figs. 4a-c), it is quite possible that the early convective activity over the lake produced an outflow that was chanelled in between the Cordillera and the Northern Serrania; so it could be that the lake breeze was susbtantially enhanced by a convectively-induced outflow.

It is interesting that the simulated reflectivites are not high for deep convective storms (Fig.4) despite the presence of hail. The authors make no mention to this finding. This result suggests that the heavy hailfall occurred because of high-terrain effect, that is, for the 0°C isotherm being very close to the ground at the elevated terrain of La Plaz. Or else, the WRF simulation underestimated the storms´ severity.

**Page 7, line 16:**
"*The lake breeze front is accompanied by strong winds at 500hPa...*"
How was the lake breeze identified? It is unusual to utilize 500 hPa winds in tandem with surface winds to characterize a lake breeze.

**Page 7, lines 18-20:**
"*We observe at the same time an intensification of previous convergence zones around complex orography; with a propagation of the convergence areas from the previous zones towards each other (Fig. 5e).*"
The magnitude and noisy character of the convergence-divergence patterns indicate that the "convergence zones" are all contaminated by ongoing deep convection in the simulation. So we are basically looking at the environment modified by the storms themselves instead of convergence as a pre-storm lifting mechanism. So I think the above analysis is a bit confusing regarding what exactly the authors wish to discuss. Terrain effects? Lake-induced circulations? The problem is that at this stage (Figs.5e-f) the mesoscale enviroment is already highly modified by the ongoing convection such that it is difficult to isolate the mesoscale forcing mechanisms based on divergence.

**Page 7, lines 31-32:**
In the analysis of the skew-T diagrams I think the authors missed a few important features: the temperatures and dew-point temperatures are very low if we consider a typical environment for severe convective storms. Naturally, this is explained by the very high elevation of the local terrain, but, still, this point deserves to be stressed since most readers are not familiar with such environments. On the one hand this aspect does not preclude the generation of CAPE (as shown in Fig.6e), but on the other hand it does reduce significantly the precipitable water, which is not informed in the text. In fact, the simulated 2m-specific humidity was rather low over La Paz (Figs.5a-c). Given these points, it is intriguing that a significant flash flood was observed in La Paz, as confirmed by the videos. So this raises a few questions:

has the rainfall alone accounted for the observed flooding or has the melting of hailstones contributed to that occurrence? Or else, is there any indication that the WRF simulation has underestimated the moisture supply for that region? Perhaps, very steep terrain leading to fast surface run-off also accounted for that event? This is an important result and discussion because most forecasters (at least the ones working in lower terrain areas) probably would not cite flash flood as a main threat if looking at those simulated/forecast soundings.

I probably would not say that the thermodynamic profile in Fig.6a is stable (line 31), but approximately moist neutral. It would not be hard to become unstable even with just some little surface heating. By the way, did WRF underestimate the 2m-temperature?

**Page 8, lines 6 and 8-9:**
"*The location of this band overlaps the lake-valley breeze convergence zone.*" & "*...responsible lifting mechanisms identified until now are orography and lake-valley breeze convergence.*"
As mentioned earlier, I think that a closer analysis of the simulation suggests that the (simulated) lake breeze was enhanced by convectively-induced outflow from previous convection.

**Page 8, lines 6-8:**
"*The evolution of the intensity of vertical velocity at 4000 meters above ground level (magl) and wind shear from surface to 6000 magl (Fig. 7a-c) gives an idea about the severity of afternoon convection and resulting storm.*"
The authors do not develop the discussion here. Were the simulated vertical velocities intense? They do not appear particularly intense to me. Even at 2 km horizontal grid spacing, simulated severe storms can develop vertical velocities of the order of 10 ms$^{-1}$. Isn´t it possible that WRF has underestimated storm severity in this case? Or maybe it is because large hail accumulation at very high terrain does not require strong updrafts typical of true supercells. Any information available regarding the observed size of the hailstones? Sometimes, at higher elevations, storms produce copious amounts of small hailstones, for which very strong updrafts are not required.

**Section 3.3:**
I think that the discussion around the results from the sensitivy analysis was rather superficial and, therefore, mostly inconclusive.

**Page 10, line 3:**
"*On 19 February 2002, surface wind over the altiplano was guided by thermal lake, mountain and valley breeze effects.*"
Again, I would say that the WRF control simulation suggests that outflow from previous convection also played a role by influencing the strength of the lake breeze (i.e., by enhancing it).

**Page 10, lines 17-18:**
"*The presence of sufficient wind shear extends and supports the organization of convective storms in terms of multicells, supercells or mesoscale convective systems*."
In other words, no conclusion was reached regarding the convective mode. Unfortunately, the authors did not check for the possible development (or non-development) of mesocyclones within the storm cells simulated in their D4 domain. That would help identifying the convective mode (as for multicells versus supercells).

**Page 11, lines 5-6:**
"*....suggests that this severe event was in fact part of a mesoscale convective system.*"
Throughout the text the authors have not presented arguments to sustain such conclusion.

---

## Referee Comment (RC2) · Anonymous Referee #2 · 12 Mar 2019

**Review of nhess-2019-27**
**"Characteristics of a Hailstorm over the Andean La Paz Valley"**
by Marcelo Zamuriano, Andrey Martynov, Luca Panziera, and Stefan Brönnimann
Submitted to *Natural Hazards and Earth System Sciences*, February 2019.
https://doi.org/10.5194/nhess-2019-27

**Summary**
On 19 February 2002 a severe storm hit La Paz Valley, causing 69 casualties. Unfortunately there are only few low-resolution observations available, mainly from satellite or from a low space-and-time resolution network of stations (reporting mainly daily accumulated rain).

Because of that scarcity of observed data, the case study is tackled using WRF simulations (with 2 km in the inner domain) and applying some sensitivity tests, to highlight the important factors that favored this severe storm. Tests include smoothing or removing orography, warming or removing the Titicaca lake and suppressing heat fluxes. It is found that all these tests affect -more or less- the results.

**Major comments**

- First of all, the authors must clear that the insights found on the mechanisms underlying the development of this storm are indeed insights on how the WRF simulations works, more than on how this specific storm really evolved. In fact, the observed data are so few, that while the authors can state that the control run has a general agreement with the low resolution satellite-derived observations, they can't prove that the local-scale features evolved following the details simulated by WRF, since there are no radar data, no lightning data and no hail data to compare with. Also, there is no clear comparison with the only available precipitation data, that is, *"The most important rainfall quantity was therefore registered around La Paz next to the mountain slopes (with measured values of around 50 mm)."* I assume that 50 mm of precipitation is the *daily* accumulated value, while the simulations data of Figure 10 show only 3-hours accumulated rainfall, with maximum peaks of about 20 mm. That is not a good forecast if what reported by Hardy (2009) is true, that is, that there were 39.4 mm in only one hour.

- While there is much emphasis on hail, from the title of the paper to the tentatives *"to investigate the physical processes leading to the hailstorm formation"*, in the presented material there is really no specific information on hail and in particular on what would favor a severe hailstorm instead of, for instance, a severe storm not characterized by much hail (that is what one would expected from the analysis of only one case study). The Hailcast module

is coupled into WRF, but I can't find anywhere any clear Hailcast result, like, for example, a map of simulated maximum hailstone diameters. Only a small blue line (very hard to note) is overlapped in Figures 4 and 10, to show hailstone "around 5 mm" or "between 4 and 6 [mm]" of diameters, which seem too few for evaluating the hailcast forecast. Lastly, how important was the hail aspect for the damaged produced by this storm and, in particular, for the 69 casualties? I suspect that the flood aspect (with saturated terrains and steep slopes) could have been more important. In conclusion, maybe that emphasis should be given more on the flash-flood aspects than on hail (and the title changed). Otherwise, the key role of the hail should be better highlighted.

- The material is not presented in a very linear form and there are many repetitions, like for example between Section 4 Discussion and section 5 Summary and Conclusions. I encourage the authors to remove most repeated material and to merge together sections 4 and 5.

- Figures are really too small and very difficult to understand. I suggest to put only one "large" figure per page. Please add at least one map of hail simulated by Hailcast.

Because of these major comments, I kindly ask for a major revision.

**Minor comments**

Abstract: *The iconic hailstorm*
"iconic" is an appropriate term? I would remove it here and during the whole manuscript.

1, 19: *taking into consideration Bolivian farmers perception of extreme events*
Is it pertinent?

1, 20: *Andean farmers perceive an increase of the frequency and intensity of storms and hail*
How much scientific can be such statement?

1, 25: *was described as an unprecedented crisis*
Maybe crisis is not the best word to use here.

2, 1: *(that resulted in 69 casualties*
How they died? Because of flood, wind gust, giant hail?

2, 5: *the knowledge is taken from local peoples perception of hail frequency and intensity*
That is too few to write a paper on "hailstorm".

2, 5-7: *The lack of formal physical process knowledge about local thunderstorms formation over this region is evident as we take as example the explanation given by the SENAMHI about the*

*plausible mechanisms for this particular cell formation.*
I don't think that criticizing SENAMHI explanation is relevant here. If their explanation *"might sound trivial for a super-cell formation"*, also your explanation that convective cells *"were triggered by a mix of low level wind convergence, surface heating and orographic forcing"* (page 10, line 8) might sound trivial to others.

Introduction: In this section I would expect some explanation about literature studying hailstorms cases, like for example the work by Kunz et al. (QJ RMS 2016), if emphasis on hail will be maintained. Also some references to orographic precipitation studies could be added, like for example seminal works from Rich Rotunno, Robert Houze, Daniel Kirshbaum and others.

3, 4: *1745 LST*
I searched in Internet for what means LST and it is the "Local Sidereal Time", while you probably was reffering to the "Local Time" (LT). Please correct all LST and describe the relation between LT and UTC time for your specific location.

Figure 3a: I can't really see the observed rainfall distribution. Maybe you can plot the accumulated values at each SENAMHI station located in that area?

4, 16: *We study the role*

4, 21: *We assess the presence of the main ingredients for a hailstorm to occur (moisture, instability and lifting)*
First of all, that language is more appropriate for a cookbook recipe than for a scientific article. Second, if there are -unfortunately- some people that oversimplify the thunderstorm forecasting problem to such a level, they do it for thunderstorm in general, not specifically for "hailstorm". So, please, describe some general features of the environmental conditions that favor thunderstorm development and, if you have any evidence, of specific conditions that instead favor hailstorm formation. Otherwise, simply list the parameters that this study investigates.

4, 23-24: *The low level moisture transport vectors were calculated following*
$IVTU = \frac{1}{g} \int_{SFC}^{200} qu\,dp$
*...It is calculated from the surface SFC up to 200 hPa.*
I really can't understand how can you call it "low level" moisture transport if it is calculated up to 200 hPa!

Figure 2a-2d: Please define what are "sensor count"?

4, 27-28: *the presence of low level water vapor is not well captured in this band but its corroborated with infra-red image at 12μm (not shown).*
Why infra-red at 12μm should provide information on low level water vapor?

6, 8-9: *We note that the models rainfall spatial distribution corresponds very well to the clouds locations in Fig. 2a-b)*
While there is a general agreement, I would not say that forecast and observation fits "very well". Can you provide any verification measure of the agreement between observations and forecasts?

6, 14-15: *Thus WRF is able to simulate the event with its most important features.*
Not so sure, also because you do not have enough observations to describe in detail the event features.

6, 20: *(with measured values of around 50 mm).*
In 24 hours? What is the WRF forecast in the same period and location? Is 50 mm in one day an exceptional rain value? BTW, what is the rain climatology for that location and period of the year?

6, 24: *The analysis of the large scale characteristics and the few observations available provides insufficient information about the three basic ingredients for a thunderstorm: moisture, instability and lifting.*
Maybe it is better to remove it, since it does not add any useful information.

6, 28: *in order to explore the chronology of the precipitation.*
"chronology" is appropriate?

7, 3: *A closer look to the maximum radar reflectivity*
A closer look to the simulated maximum radar reflectivity.

7, 6: *even hailstones of around 5 mm are simulated at the centre of the two formed cells.*
Is it the Hailcast maximum diameter? Or the mean diameter? How it compares with observed hailstone diameters (e.g. from media report) and with the locations were hail was reported? Why not showing also a full map of hail diameters as simulated by Hailcast?

7, 11: *comes from the Amazon avoiding the cordillera obstacle*
"avoiding" is the appropriate term?

Figure 5a-c: Reference vectors are 50 [kg m$^{-1}$s$^{-1}$] ?

Figure 5d-f: Reference vectors are 3 [ms$^{-1}$]?

7, 29: *in the proximity if the rain-band*
in the proximity of the rain-band

8, 2: *the atmosphere is saturated until 400 hPa with important wind shear favoring hail and graupel formation*
Can you add a reference to support the hypothesis that wind shear favor hail and graupel formation? In figure 6f I can see a strong directional shear at about 500 hPa, but for example also in figure 6e

there is a strong directional shear at about 550 hPa, even if wind directions are completely different. So, the important issue was really the wind shear or rather the wind direction and intensity absolute values?

8, 7: *and wind shear from surface to 6000 magl (Fig. 7a-c)*
How you define wind shear? Is it the "bulk shear"? I.e. taking the magnitude of the vectorial difference between wind at 6 km and wind at 10 m? Please explain.

8, 13: *Nevertheless, convergence is not enough to explain deep convection.*
Is that a scientific explanation?

9, 9: *showing that breeze and orographic lifting are enough for producing rainfall.*
Is that a scientific explanation?

9, 17: *leaving the cordillera storm free with isolated hailstorms*
Sorry, can you rephrase?

10, 6: *following the thermo-topographic circulation.*
Can you explain better?

10, 13-14: *This propagation allowed both cells to join each other resulting in a precipitation band. This auto-propagation mechanism has been observed*
Please, can you explain better?

10, 17-21: *The presence of sufficient wind shear…as shown in Fig. 7f.*
This part seems not much relevant with the experiments described. I suggest to remove it or to move it to Section 3.

11, 5-6: *satellite information and reanalysis suggests that this severe event was in fact part of a mesoscale convective system.*
That option was never analyzed nor mentioned before the Conclusions. Please remove it or discuss it with supporting facts already in the previous sections.

11, 20-21: *And the surface heat flux suppression (NOHEAT) highlights the importance of surface energy fluxes for atmospheric instability.*
Isn't it a trivial result?

11, 22-23: *highlights the complex interaction between large scale circulation, orography and local features in the formation of hailstorms over the tropical Altiplano.*
Sincerely, I have not found any specific information that can explain why a hailstorm was formed, instead than a thunderstorm (or supercell) not particularly characterized by hail.

11, 23-24: *A semi-comprehensive scheme of participating mechanisms can be found in Fig. 11.*

Please, explain how figure 11 describes this (participating?) mechanisms in details or remove this figure.

11, 29-30: *the proposed mechanisms of this hailstorm formation should be confirmed by high resolution observations*
You already said that these observations are not available.

Regards.

---

## Referee Comment (RC3) · Anonymous Referee #3 · 14 Mar 2019

**Review „Characteristics of a hailstorm over the Andean La Paz Valley"**
**by Marcelo Zamuriano, Andrey Martynov, Luca Panziera and Stefan Brönnimann, submitted to NHESS**

The manuscript examines a severe convective storm event associated with large hail and heavy rainfall that occurred in February 2002 over the La Paz region in South America. Due to the scarce availability of suitable observational data, WRF simulations complemented by sensitivity studies with different model setups were conducted. The main objective of the study is to identify the mechanisms and processes most relevant for the triggering and maintenance of the convective storms. The topic of the paper is basically relevant for NHESS. However, there are a number of important issues that have to be considered and addressed before the paper can be published. Above all, an in-depth discussion of all figures, the findings and the most important results is necessary.

All my suggestions and comments are listed below as major, minor, and (a few) editorial points.

**Major revision points:**

1) The results and illustrations are mostly discussed very briefly, superficially and descriptively. A thorough, in-depth discussion and interpretation is lacking. This applies in particular to the sensitivity studies and to the application of the Hailcast model. In order to use them in a sensible way, a much deeper discussion and interpretation of the results is necessary. Furthermore, the interconnection of all subsections has to be improved.

2) Please explain in which way the result can be generalized and how the findings may apply to comparable events.

3) I do not agree that the model capture well the main features of the severe thunderstorms. There are large discrepancies in both the location and precipitation intensity. Even the two major convective cells are not well reproduced by the model. The conclusion (Sect. 3.1.2) that the model is able to simulate the main features and, thus, capture the mechanisms decisive for the triggering and maintenance of the storms is not justified.

4) The effect of a lake breeze that is relevant for convection triggering cannot be derived solely from the wind field, but requires analyses of temperature / moisture gradient or vertical lifting. As sea/lake breezes have a limited vertical extent of a few hundred meters, they cannot be detected at 500 hPa, not even over the elevated terrain around Lake Titicaca.

5) The discussion Section (together with the Conclusion Section) is more a summary than really a discussion (this applies at least for the first half of the text). I'd like to see a more thorough discussion and a better synthesis of the results instead of repeating what was already written. I also suggest to extend the comparison with other related studies and references.

6) **Be accurate in your citations**!
Sect. 2.1.2: I'm puzzled about the statement "known uncertainties of precipitation estimates over complex terrain (Rasmussen et al., 2013)" as this does not make sense for the TRMM algorithm. The cited reference investigated the range of the rain bias in storms containing four different types of convection in extreme radar echoes over South America, but did not investigate a relation between bias and terrain characteristics.
The study of Kunz et al., 2018 investigates a supercell over low-mountain ranges and not over the Alps.
These are just two examples that I've checked.

7) The number of cited references is rather small.

8) Most of the Figures are too small and/or have a too low resolution; it's hardly possible to see any details. Furthermore, as most of the readers are not familiar with the study area, an

additional figure that indicates all areas and cities referenced in the text (Altiplano, Amazonas, …) would be helpful (Fig. 3a is too small!).

9) The English language requires a thorough check by a native speaker. I have listed a few corrections, but these are too numerous to fully list.

**Minor points**

1) The title should be changed as the paper goes beyond the investigation of one single hailstorm.

2) Abstract: Can you highlight the most important findings (e.g., We show the importance of orographic configuration… is too general; be more specific). Consider to move the last sentence after the 2nd one.

3) The use of LST is weird: Twelve noon local solar time (LST) is defined as when the sun is highest in the sky, i.e., it is a function of the geographical coordinates. Do you mean local time instead (LT)? What is LT in UTC at La Paz?

4) Introduction, 1$^{st}$ paragraph: Give a brief reason why hail hazard assessment is not available. But also note that there are some studies available that uses either satellite data (e.g., Cecil and Blankenship 2012) or reanalysis (e.g., Prein and Holland 2018) to estimate hail frequency, that may be mentioned (e.g., by giving an estimate how frequent the region is affected by SCSs, similar events in the past, …).

5) Introduction, 2$^{nd}$ paragraph: give some more details about the event: total rainfall accumulation, hail sizes, wind gusts; what was the reason for the large number of casualties?

6) P2L5: "The lack of formal physical process knowledge" is unclear: I think you mean what makes this region special related to thunderstorms compared to others? The physical processes for thunderstorm formation are the same all over the world.

7) Introduction, last paragraph: Say a few more words about the motivation and objectives of this study to be published. Only saying the goal is to better understand the processes is too simple. What are the research questions / hypotheses? What are the reasons for performing sensitivity studies?

8) P3L14: how many gauges? If all stations are shown in Fig. 3a, you cannot say "around 1 km distance" given the large spatial differences.

9) P3L16: "some data quality issues": what do you mean?

10) Sect. 2.2.1, 2$^{nd}$ paragraph: give some more details about the schemes and the configurations you have used (besides: write out YSU); also give a reference for the radar-forward operator you have used to create Fig. 4. Furthermore, you should motivate here - or maybe better in the introduction (see minor point 7)- the reasons why you have conducted the sensitivity experiments. What do you expect from those additional runs?

11) 2.2.2: Why did you not consider deep layer shear (DLS) or storm-relative helicity SRH, which are important ingredients for supercells? The unit of IVTU/IVTV in 100 g (s m)$^{-1}$ (yes, this is the result when using the units you stated) is very strange and not an SI unit. I strongly recommend to insert dp not in hPa, but in Pa. Why do you integrate until 200 hPa?

12) Always put a blank between two units (e.g., ms$^{-1}$ could be m/s or millisecond).

13) How is CAPE computed? Mixed layer CAPE, most unstable CAPE, …?

14) P5L4-6: As you did not investigate runoff or flash floods, soil saturation is irrelevant. You may mention this in the introduction, but not in the result section.

15) Sect. 3.1.1 / Fig. 1b: Showing only one isoline of the geopotential at 200 hPa (also without labelling) does not make sense as the flow at lower levels, which are most important for the triggering, could be completely different from that at 200 hPa.

16) P5L13-14: what do you mean by "northward displacement of the Bolivian High"?

17) P5L18: specify what is meant by "mesoscale features"

18) Fig. 2: There is a discrepancy between blue shaded contours and the color bar at the bottom (besides: the numbers are illegible).

19) P5L24: "…show a remarkable spatial consistency…" this statement is too general and too optimistic; of course, there is some consistency (mainly at 17 LST), but also some discrepancies. This is what one would expect as you compare visible cloud areas with rainfall.

20) P5L25: "Titicaca lake, the Amazon region, and the eastern cordillera…" These regions / features should be indicated in one Figure.

21) P5L28-30: "important convection…shallow convection". From the visible channel solely, you cannot distinguish the intensity of convection; "TRIMM is not able to capture any light rainfall". Why do you suppose that rainfall already started at that time?

22) P5L7: Why do you consider longwave radiation here? Explain and motivate this (and recall that Figs. 2a-d show the visible channel).

23) P5L10: The Figures show radiation and precipitation, and not water vapor as stated here.

24) P6L25: Also consider vertical wind shear.

25) Sect. 3.2.1, 1$^{st}$ paragraph: So what? What can you conclude from the Hovmoeller plot?

26) Sect. 3.2.1, 2$^{st}$ paragraph (see also major comment 4): The strong north-westerly flow east of the lake between 11 and 13 LST can also originate from the cold pool of the convective cell. This may also explain the large flow divergence at 13:00 around that cell. Note that sea/lake breezes have a rather small vertical extent so that the associated wind field is of minor importance for the movement of the convective cells.

27) Fig. 5a-c: the color code of the figures and the color bar do not match; what are the areas indicated by the brown color?

28) P7L10: "While the surface humidity follows the lake breeze…" I cannot see any relation.

29) Sect. 3.2.2, 2$^{nd}$ paragraph (see comment 26): sea / lake breezes typically have a vertical extent of a few hundred meters. They cannot be identified at 500 hPa. Even though at 14:00 there seems to be a frontal boundary involved, which was not the case at 11:00 and even at 12:30. In the previous paragraph, however, you suggested the wind field even at 11 and 12 is associated with the lake breeze. Furthermore, I'd suggest not using different times for the plots (12:30 in Figs. 5-6 is not shown by Fig. 4). Finally, relate the fields to the convective systems.

30) Fig. 6: It's hard to see the CAPE. I suggest to include additional Figures showing only CAPE. Furthermore, it's not clear what Figs. 6a-c show.

31) P7L24-26: It's rather confusing to term that sensible heat is released as the Figures show the vertical heat flux; Solar radiation is surely not the only reason for increases in CAPE (cf: the largest increase in CAPE in the northeast is associated with lowest sensible heat flux.)

32) P7L30: As the city heat-island effect is not relevant for the convective storms, I suggest to omit this statement.

33) Fig. 7/8: It would be easier for the reader to show shear as amount and not as vector.

34) Sect. 3.2.4: In some places reference is made to convection triggering. However, convection on that day is triggered earlier as shown in Fig. 4. Be sure what is really meant here: maintenance, triggering of new cells, or – as you describe – the merging of scattered convection to a larger band showing some basic features of a squall line.

35) P8L12-14: Explain where the cold pool is located (I cannot see a convergence line; besides, the cold pool cannot be situated over the convergence line). It's not possible to speculate about a supercooled state without showing hydrometeors.

36) P8L21: This explanation for a cold pool is wrong; rather evaporation and sublimation cooling by hydrometeors drives the cold pool.

37) P8L28-29: Literature?

38) P8L7: "convection is present without thermodynamic instability" this is a contradiction in itself.

39) P9L13: How is hail formation suppressed? Changes in microphysics, intensity/size of the updraft, or what else?

40) Experiment with no lake: when you state that the wind field for this realization is similar to a lake breeze, than I doubt even more that really a lake breeze is responsible for CI.

41) Caption Figure 10: Make clear that the smaller "hailstones" with a diameter of 4 mm are treated as hail in WRF only; according to WMO definition, hail has a minimum diameter of 5 mm.

42) Discussion: All the above critic points likewise apply to the Discussion Section

43) P10L13: Cold pools behave as density currents. Thus, their propagation results from the interaction between mean flow and density current. They do not directly propagate with the mean wind. Furthermore, cold pools emerging between two cells leads to flow divergence, which prevents and not favors cell merging. The term "auto-propagation" is not appropriate here.

44) P10L17-18: Low to moderate wind shear does not allow for substantial hail formation as it also affects the strength / width of the updraft (cf. Dennis and Kumjian, 2017 in JAS).

45) P10L24: The conclusion about a relation between instability and surface fluxes are not justified as the largest increases in CAPE occur in a region with lower fluxes (NE parts).

46) P10L26: The reduced temperature gradient was not shown.

47) Section 4, last paragraph: The question of a possible trend in the frequency of hailstorms is irrelevant for a single case study.

48) Conclusions: The most important results should be more clearly identified and highlighted. Clearly show what is new, what the reader should have learned from the study.

49) P11L6: Usually an MCS is defined as an ensemble of thunderstorms that produce a contiguous precipitation area on the order of 100 km or more in horizontal scale in at least one direction (e.g., AMS glossary). This definition does not apply to the 19 Feb. storm.

50) P11L6 (blocking): You haven't shown that blocking really occurred on that day. This, of course, would increase the substance of the paper.

51) P11L23-24: I suggest to move this part to the discussion section, but also to explain in detail Figure 11.

**Typos / Small corrections**

1) p1L1: "iconic" is not an appropriate expression

2) p1L3: Satellite observations suggests; that **develop into** deep convection

3) p1L5: suggest**s**

4) p1L5-7: the statement about instability is trivial as this is prerequisite for convection

5) p1L6: what is meant by "rainfall discharge" as these indicate two different things?

6) p1L8:  deep convection

7) p1L25: that  between

8) p1L25: change "presented" by "designated"

9) P2L19-20: with numerical **model** studies

10) P3L9: "…low weather station density…"

11) P3L12: specify what is meant by "The nature of the event…"; replace "demands" by "requires"

12) P3L16: affect**s**

13) P3L22: consider to change "physical" into "main triggering"; change "system" into "model"

14) P3L30 and elsewhere: **Kain**-Fritsch scheme and not **Kein**-Fritsch scheme

15) P4L7: for all set**s** of…

16) P4L8: Begin a new sentence as the content changes: "…February 2002). They are useful…"
17) P4L16: "We study **here** the…"
18) P5L1: "…is assessed by  the convective available…"
19) P5L18: delete "still"; "…how **often** (or **frequent**) this…"; what do you mean by "formal classification"?
20) P5L28/38: the phrases "important convection" or "important cells" are weird
21) P6L1: You changed the tense within the same sentence
22) P6L4 "The infra-red images are almost the same as the visible channel" This general statement cannot be true; please formulate clearly what is meant.
23) P6L5: again avoid "important"; rather name the intensity
24) P6L19: delete therefore
25) P6L20-21: I'd suggest to slightly change the sentence "…which reflects the expected spatial heterogeneity of the convective precipitation"
26) P6L21: "In consistency with…" or "In accordance.   "; confirm
27) P6L25: "After  confirmation…"
28) P6L28: don't start a new subsection with "We therefore…"; delete the brackets (Fig. 3a)
29) P7L3: "A closer took **at**…"
30) Caption Fig. 4 (also applies for other positions): change curly vectors → wind vectors; and give a reference (arrow length vs wind speed).
31) Fig 5: indicate the unit of the reference vector
32) P7L15: "…conditions **at** 1100…"
33) P8L5: propagate
34) P9L1-2: what does it mean: "rain region is less organizes"; "hail originated by valley breeze…" sloppily expressed (hail originates from clouds); you mean the triggering of convection?
35) P9L4: replace exists by occurs; L5: can still be → nevertheless is still
36) P9L6: correspond
37) P10L9: grew in **size**
38) P10L19: they → it
39) P10L25: show

---

## Author Comment (AC1) · 15 May 2019

**Reply to Referee 1 regarding the Manuscript NHESS-2019-27: "Characteristics of a Hailstorm over the Andean La Paz Valley", by M. Zamuriano et al.**

OVERVIEW

We thank the referee for the detailed revision of the manuscript that helped us improve its quality. We have taken note of the comments and we would like to follow them with our answers. We have enumerated the comments and accompanied them with our replies marked with R:

TITLE:
1) The title makes reference only to the hailstorm, but throughout the text the authors also mention/discuss the occurrence of a flash flood accompanying the same weather event. Therefore, the title should be modified, perhaps by reading "Characteristics of a Severe Convective Storm over...". The main point is that the authors give equal emphasis to the hail precipitation and to the flash flood (rainfall amount) in the text, but the title, as it is, does not reflect that.
- R: It is true that casualties during this hailstorm are related to the flash-flood accompanying this event. While we concentrate our research on the atmospheric characteristics of this hailstorm, we acknowledge the role of the flash-flood in this natural disaster and we therefore agree that the title can better reflect this. A title that better summarize the outcome of the research would be: y"Numerical Insights of a Severe Convective Storm accompanied by Hail and Flash-flooding over the Andean La Paz Valley". This is not final and can be still changed in the revised manuscript.

FIGURES:
2) Most figures are appropriate for describing the results, but they are way too small, making it hard to read and to verify several of the important detailed information discussed in the text. It is true that the digital file allows for the zooming in of the figures, but this is rather cumbersome for the reviewer. If, for example, the authors keep one figure per page (in the submitted version) then the figures can be enlarged.
- R: We agree that the figures can be hard to read in its current form and we now include clearer, one per page, figures.

3) Captions can be improved and/or do not provide full information of the contents of the figures:
Caption of Fig.1a must inform the horizontal grid spacing for each domain.
Caption of Fig.2 begins with"(a)-(d) Remote Sensing observations assessment" which could be replaced by "(a)-(d) GOES-8 visible imagery (grey shading) and TRMM estimated 3-hour accumulated precipitation (blue shading)..."
Caption of Fig.3: "accumulated" instead of "cumulated".
Caption of Fig.4: Should read: "...maximum simulated radar reflectivity in domain D4..."
Caption of Fig.4: Should read: "...blue contour encloses areas with simulated hailstones equal to or larger than 5 mm in diameter...", and must inform at what vertical level this is valid. Surface level?
Caption of Fig.6: Should read: "...most unstable CAPE..." instead of "...maximum CAPE..."
- R: Captions for Fig.1a now contain grid spacing for each domain; the proposed captions for Fig. 2, Fig.3 and Fig. 6 make sense and have been modified; and captions for Fig.4 include now the vertical level of the hailstones simulated by HAILCAST (surface level)

INTRODUCTION:
4) Page 1, line 24: "...between 1420 and 1545 LST [...] a hailstorm affected the city of La Paz." Please, inform the corresponding UTC times as well. Has the hail precipitation lasted for 1 hour and 25 minutes over La Paz? That would be highly unusual; a trully extreme event. Or was the accompanying flash flood that lasted for such a long period?
- R: Since our main findings are related to thermal daytime circulation, we only considered to include times in LST format to give an idea of the local thermal context. However, we agree that the use of UTC times can be useful for many readers and we include them it in the revised manuscript. We also clarify that the precipitation duration was reported to last 1 hour 25 minutes with a peak including hail precipitation of about 20 minutes duration.

5) Page 2, line 11: "...generated a super-cell over the city (Soruco, 2012). This explanation might sound trivial for a super- cell formation..." As for a supercell being reponsible for the hailstorm, it

surprises me that the authors of this study run a fairly high resolution simulation of the convective storms with WRF but do not verify whether any of the simulated cells developed a mesocyclone. That could provide additional evidences for the supercellular nature of the storm(s). The authors should look for such evidences in the 2 km grid-spacing simulations through the analysis of convective updrafts correlated with (negative) vertical vorticity. More detailed comments on that matter follow below.

**- R**: We realize this paragraph in our manuscript is not very clear and does not convey our main message. We intended to point to the lack of knowledge of this kind of events by citing the supercell explanation given by the SENAMHI without any formal evidence. This study doesn't aim to study the super-cellular nature of the storm and we purposely left the mesocyclones out of the analysis. However, this comment raises a very interesting question that can be addressed in further investigations.

DATA AND METHODS:
6) Page 2, line 25: "...a temporal resolution of 6 hours and a spatial resolution of around 0.75 x 0.75 lat-lon..." I am not sure that we can state that the temporal resolution of the ERA Interim is of 6 hours since we would need at least 2 "time-steps" (i.e., 12 hours in this case) to minimally resolve any atmospheric feature using this dataset. The same comment holds for the spatial "resolution". I suggest rephrasing by "...the gridded data is available at 6-hour intervals..." and by "...with horizontal grid spacing of 0.75° x 0.75° latitude-longitude...".
**- R:** The referee is right about ERA-interim and the terms used in our manuscript. We have put effort to rephrase this paragraph.

7) Page 2, line 26: "...geopotential fields at 200 hPa, and specific humidity and winds at 500 hPa..." The authors extract 200hPa geopotential fields from ERA Interim but never show these fields explicitly. It must be indicated what is the above-ground height of the 500 hPa pressure level over La Paz. This is important because, at first, it sounds strange to analyze the 500 hPa humidity fields when we should be mostly interested in the analysis of the low-level moisture (below 3000 m AGL). It turns out, however, that La Paz is situated in very high terrain and therefore the 500 hPa fields may represent the (local) low-troposhere, which is unusal for most regions.
**- R:** We use the 200 hPa geopotential field for synoptic conditions analysis and it is included in Fig.1b. However, the referee is right that we do not show any values and we use it only qualitatively for identifying the position and intensity of the Bolivian High; the revised version adds a comment about the intensity of this field with the correspondent values. Concerning the 500hPa pressure level, the referee makes a fair point that this pressure level can look strange without stressing that it reflects low-level circulation in this particular region. We agree it is important and we do better justify the use of this pressure level in the revised manuscript.

8) Page 3, line 8: "...they provide area-wise estimates with a fair temporal resolution..." I would rather state more explicitly that the 3-hr sampling interval from the TRMM satellite, despite not being adequate for monitoring the evolution of a single severe convective storm, is the best available remote sensing data for this specific case study. The authors only utilized the rainfall estimation product from TRMM satellite. Given the severity of the storm, other products could have been analysed, such as the height of the 40dBZ radar reflectivity just as one example. South American hailstorms are known for being very tall, particularly in the La Plata Basin sector. Most readers will be curious about the depth of this cell in Bolivia; has TRMM sampled the storm at its mature stage?
**- R:** We have looked to satellite radar data (product 2A25 from TRMM) and unfortunately the radar missed the event (because of the trajectory offset), so we have added a sentence about this. We also agree that the nature of this event (very high altitude) raises some questions about how it can be compared with other regions, for example in relation of the cell depth; we are happy to add this information (from simulations) in the revised manuscript.

9) Page 3, line 14:
"...resolution network of rain gauges; the network is maintained by SENAMHI."
Is this an automated surface network? Or is it manned? This must be informed for the sake of completeness.
**- R:** This information is now included.

10) Page 3, line 25:
"...over the Bolivian central Andes D1, D2, D3 and D4 of 54, 18, 6 and 2 km of grid size..." I wonder if the D1 domain with 54 km horizontal grid spacing is really necessary when downscalling from ERA Interim. The downscale "leap" from ERA Interim directly to the 18 km grid spacing may had sufficed. Any comments on that? Please, provide the number of gridpoints (matrix size) of the 2 km mesh.

**- R:** It is true that an outer domain of 18 km may have sufficed for the simulations described in our manuscript. However, as a pre-test we contrasted results using ERA-Interim and FNL analysis (GFS based at 1x1 deg resolution) as initial and boundary conditions, and we designed a horizontal grid configuration compatible with both datasets (we considered a leap from 1 deg to 18 km grid size too big). We have kept this configuration because we considered the results were good enough and taking out the outer domain wouldn't change the main findings. If we were to propose an operational forecast configuration, we would take out the outer domain. We added a comment on this alongside with the number of gridpoints of the finer domain.

Page 3, line 30:
11) Mispelling: "The KAIN-Fritsch scheme...."
**- R:** Corrected

Page 4, lines 1-2:
12) "The initialisation time is fixed to 1400 LST on 17 February 2002, allowing enough spin-up time until the event." First question: 14:00LST = 18:00UTC? I understand the authors´ concern with the model´s spin-up period but, in my experience and from several other numerical studies on convective storms, initializing the simulations 24-hr before the convective event usually suffices for that matter. Starting 48-hr in advance (as done here) may lead to too long a "forecast range" to produce the best possible simulation. Have the authors tested distinct initialization times for the simulations? If so, was the choice of utilizing the one starting 48-hr before the event justified for being the simulation with best correspondence with observations?
Finally, were all four domains initialized at the same time? These pieces of information should be informed.

**- R:** Yes, 14:00LST = 18:00UTC (we have proceeded as responded to comment 4). Also, we have tested different spin-up times initialized at the same time in all domains. We agree that we could have used a shorter spin-up, but we have found a good correspondence with satellite data with this configuration with respect to the location and timing of the main cells (Fig.2c,g). We do realize Figure 2 did not t reflect that, so (as stated in comment 2) we have updated this figure.

Page 4, line 20:
13) The authors have available the output of a fairly high resolution WRF simulation (their domain D4) of the convective storms, but as "hailstorm diagnostics" they follow an ingredients-based approach ("We assess the presence of the main ingredients for a hailstorm to occur...") for which having a high-resolution simulation is not indispensable. I recognize the importance of the ingredients-based approach, but additional diagnostics should have been chosen that explore the full explicit information made available by the high resolution simulations. Interestingly, in the Results section, the authors do show variables/fields such as simulated reflectivities, updraft strength, surface winds, and areas enclosed by hailstones surpassing a given diameter threshold, but none of these variables/fields is mentioned in the methodology as a diagnostic. The parameter "updraft helicity", computed around 3 km A.G.L., would be also a natural choice of diagnostic to verify if the simulated storm(s) displayed mesocyclones (i.e., if they behaved as supercells) in any given stage of its(their) development. At least, vertical velocities should be analyzed in tandem with vertical vorticity in order to assess the presence (or the lack thereof) of mesocyclones. Surface winds/outflow produced by the simulated storms are shown in the Results section but could be better utilized by the authors when assessing the storms´ severity. Finally, the presence of moderate to strong vertical wind shear is among the typical ingredients for severe convective storms, but the authors do not include any parameter for vertical wind shear in this section, despite discussing this parameter in the Results section.

- **R:** We acknowledge that this section must be updated with the variables used in the Results section and it has been done. The non-use of the updraft helicity parameter was discussed in the reply to comment 5. We agree this parameter could be interesting for future research. In the meantime, we propose to discuss the values that we found in order to assess the severity of the storm, since its location has good spatial correspondence to the updrafts position in Fig.7c. We

also have updated this section for a better introduction of the wind shear and the rest of parameters used in the results section.

RESULTS:

14) Page 5, lines 9-10: "...the well known anticyclone at 200 hPa (also called Bolivian High) was located over the north-east part of Bolivia (Fig. 1b)."
How the Bolivian High was characterized? The authors do not show the 200 hPa geopotential heights in Fig.1b. To a large extent, the Bolivian High is a - R to the intense convective activity (latent heating) observed over central South America during the warm season, so it is as much a consequence from deep convection than the cause for it. The discussion in Section 3.1.1 indicates that the Bolivian High drives/influences the convective activity but does not stress the important feedback from the convection itself.
**- R:** This is correct. But we have decided to focus in our study mainly on the influence of the Bolivian High position and intensity on the enhancement (southern position) or suppression (northern position) of moisture transport towards the Altiplano and less on the feedback. The referee makes a fair point that we could discuss the feedback from the convection itself but that would be outside the scope of this manuscript, which has been updated in the introduction section. Fig.1b is improved in the revised version.

15) Page 5, lines 15-16: "We find a considerable amount of water vapour over the Bolivian Altiplano due to the continuous precipitation episodes registered during precedent weeks." Shouldn´t the presence of water vapour over the Bolivian Altiplano be the cause for the precipitation events rather than a consequence from it? If the Amazon Basin was not the moisture source for the Bolivian Altiplano (as stated by the authors in lines 11-12 of page 5), what was the effective moisture source? I know the authors discuss this matter in more details later on in the text, but my point here is that the general perspective provided by Fig.1b alone does not convince the reader that the Amazon Basin was not a moisture source for the Bolivian Altiplano. Fig.1b also suggests the presence of the South Atlantic Convergence Zone; do the 850hPa fields (not shown) also depict that?
**- R:** We understand the referee's point about our text formulation. We concede we can not assert the moisture source merely from Fig.1b. We have reformulated this sentence.

16) Page 5, line 22: "Satellite images from GOES-8 describe the fast development of the hailstorm..." Figs.2a-d per se do not allow the identification of the hailstorm. Maybe an arrow could be superimposed to the image to indicate which cell is the hailstorm; or else the figure caption should inform that.
**- R:** Figure improved.

17) Page 5, lines 26-27: "...the presence of low level water vapour is not well captured in this band but it's corroborated with infra-red image at 12 μm (not shown)." I do not agree with this specific statement. The thermal infrared imagery at 12 μm is useful to detect clouds and storms with tops at distinct heights, but not to detect low-level water vapour. In fact, it is hard to detect low-level water vapour from the geostationary satellite imagery, with the most reasonable choice (with GOES 8) being at mid-levels utilizing the 6.48 μm channel ("water vapour channel"). As for the 12 μm channel, was the hailstorm exceptionally deep for the Altiplano region (as inferred from the brightness temperature)?
**- R:** The referee is right that this band is not useful to detect low level water vapour, but rather surface temperature and moisture. Since the results are not much different to the water vapour band (6.5 μm channel) we opted to use the 12 μm channel to confirm the soil moisture and saturation that could have played a role in the flash-flood. We have added figures from both bands (water vapour and infrarred) to the appendix and rewritten this paragraph in order make it clearer. We do keep the visible band results in the manuscript's improved Fig.2

18) Page 5, line 30 and page 6, line 1: "...with two important cells captured by TRMM at the east of lake Titicaca and surrounding La Paz city." Again, it is hard to identify these cells in Fig.2c. The authors should try to superimpose arrows to the satellite imagery to highlight the convective cells being of most interest.
**- R:** This suggestion has been appreciated and applied to Fig.2

19) Page 6, lines 1-2: Here the authors mix two verb tenses "northern cell was" and "southern cell is". Please, choose one verb tense when describing the event and stick to it throughout the text.
- **R:** Thank you for pointing out this unfortunate mix. We took more care to the manuscript tenses all over the revised version.

20) Page 6, line 3: "At this point the infra-red images are almost the same as the visible channel (not shown)." I cannot understand what the authors mean by this statement. It is best to remove it since it is confusing and does not add relevant information.
- **R:** We realize this is confusing and this is related to the comment 17. Following our reply, we decided to show this in the Appendix and make this sentence clearer.

21) Page 6, lines 3-4: "...the convective cloud development arrives to its term during late afternoon (Fig. 2d)." I would suggest rephrasing to "...the demise of the convective activity occurred during the late afternoon (Fig.2d)."
- **R:** We are glad to use your suggestion, it improves the sentence.

22) Page 6, lines 7-8: "Morning is characterized by high water vapour content and disperse rainfall." The simulated radiance from WRF does not inform "water vapour content", but provides a simulated image from the thermal infrared band which is utilized to detect brightness temperatures from distinct surfaces and cloud tops, implying (in the case of clouds) the presence of hydrometeors and not simply water vapour.
- **R:** We are aware the outgoing long-wave radiation contains only the long-wave spectra and does not include exclusively information about water vapour content, but all the thermal radiation to some extent. Nonetheless, recent studies (Sicart et al. 2015 and Sulca et al. 2018, among others) have shown the usability of OLR for cloud cover analysis over this region. We have rephrased this sentence and added the references.

23) Page 6, lines 8-9: "...the model's rainfall spatial distribution corresponds very well to the clouds locations in Fig. 2a-b over the Altiplano and cordillera, and less over the Amazon." It seems clear to me that the simulation overestimated the cloud cover/rainfall to the east of Lake Titicaca and over the Altiplano and Serranias. Moreover, the strongest simulated cell at 0800LST (Fig.2e) was located south-southwest of the respective observed cell (Fig.2a). I generally do not expect the model to nail down the exact location and timing of the convective storms, but I do not agree with the statement that "the model´s rainfall spatial distribution corresponds very well to the clouds locations"; in fact, the misplacement of the strongest cell at the early stages of the weather episode may explain some of the surface features displayed in the following figures and should be stressed in the text.
- **R:** The referee makes a fair point regarding Fig.2a and Fig. 2e. We have updated Fig. 2 and rephrased this sentence in order to support better our findings.

24) Page 6, lines 9-10: "Early afternoon (Fig. 2g) shows important water vapour at the northern cordillera..." Again, Fig.2g does not show water vapour. If the authors wish to describe the behaviour of the atmospheric water vapour in the simulation then they must plot the simulated water vapour mixing ratio (or specific humidity), not the simulated outgoing long wave radiation.
- **R:** This comment follows comment 22) and we have updated this section to make it consistent to our reply to comment 22)

25) Page 6, lines 12-15: In this paragraph the authors jump into two conclusions without presenting solid arguments to back them up. First, that the hailstorm was mainly induced by mesosale features, and, second, that the cordillera blocked the moisture flow from the Amazon. At this point they can only hypothesize these two aspects. The authors should fisrt describe the WRF simulations in more details before presenting these conclusions.
- **R:** We might have been adventurous in our fast conclusions and we accept the referee's comment. We have therefore opted to to better develop and explain the reasoning behind this conclusion.

26) Page 6, Section 3.1.3: I think the discussion in this Section is poor. First, there is no figure illustrating the analysis; second, the authors should provide more specific/detailed information instead of just stating "Some places registered no precipitation..." and "...station observations confirms that an important quantity of rainfall fell down close to complex orography...". Around La Paz

there is more than one important orographic feature, so what exactly "complex orography" are we talking about? How did the WRF simulated rainfall compare to the observed rainfall from the rain gauges? How did the TRMM-estimated rainfall compare with the rain gauges? These are relevant information since TRMM was also used to evaluate the WRF simulations in the previous section. In line 17 it should read "accumulated precipitation".

- **R:** As for comment 25, we might have considered Fig.3a self-explanatory and we didn't speculate too much about the orographic features' description and raingauges information. We have taking this comment into account, have reformulated accordingly the section.

27) Page 6, lines 24-25: "The analysis of the large scale characteristics and the few observations available provides insufficient information about the three basic ingredients for a thunderstorm: moisture, instability and lifting." Actually, the limitation goes beyond the ingredients-based analysis: the pieces of information analyzed thus far could not provide any insight regarding the internal strucuture of the convective storms, especially in the absence of a local weather radar. This is a particularly relevant issue considering the study of a severe hailstorm. Therefore, a (good) high resolution numerical simulation is desirable to provide such an important insight.

- **R:** Indeed, the limitations for this kind of study are very well summarized by the referee and we have taken into account this comment by stressing these issues in our revisioned manuscript.

28) Page 6, line 28: Avoid starting this paragraph with "We therefore...."
- **R:** Yes, we see your point and we have changed this.

29) Page 7, line 3: "A closer look AT the maximum SIMULATED radar reflectivity..."
- **R:** Thank you, we have corrected this.

30) Page 7, lines 3-6: "...reveals late morning convection in places where lake and/or valley breeze encounter complex orography (Fig. 4a)." I do not think the simulated radar reflectivity alone provides this information. Perhaps the simulated surface winds combined with the simulated reflectivity can indicate that, but, still, it is hard to reach a conclusion from Fig.4 alone. To better characterize mesoscale fronts (such as breezes) it would be better to plot the divergence fields at the low levels and look for linearly-oriented convergence features. "Later on, the lake breeze becomes more intense and pushes the rain spots towards 5 the east (Fig. 4b- c)." Given the highly divergent pattern of the simulated winds over the Titicaca Lake associated with earlier convection (Figs. 4a-c), it is quite possible that the early convective activity over the lake produced an outflow that was chanelled in between the Cordillera and the Northern Serrania; so it could be that the lake breeze was susbtantially enhanced by a convectively-induced outflow. It is interesting that the simulated reflectivites are not high for deep convective storms (Fig.4) despite the presence of hail. The authors make no mention to this finding. This result suggests that the heavy hailfall occurred because of high-terrain effect, that is, for the 0°C isotherm being very close to the ground at the elevated terrain of La Plaz. Or else, the WRF simulation underestimated the storms' severity.
- **R:** We were perhaps too intrepid to assert the convection locations from Fig.4 alone. We recognize that further analysis is needed to arrive to this claim and we decided to discuss the convection after analysing Fig.5. The referee also makes good observations regarding the possibly breeze enhancing by early convective activity over the lake, so we have added some comments on that. We also share the referee's interest to the "low" reflectivity values and we have made some comments about the freezing level (0 deg isotherm) in Page 8, line 15 and Page 11, line 14. We nonetheless agree with the referee that more can be said about these findings and we have underlined better this feature in the revisions.

31) Page 7, line 16: "The lake breeze front is accompanied by strong winds at 500hPa..." How was the lake breeze identified? It is unusual to utilize 500 hPa winds in tandem with surface winds to characterize a lake breeze.
- **R:** We realize we have mixed lake breeze (using surface wind) and 500hPa wind circulation. The main point of this is to start to relate surface to low level circulation for the purpose of introducing wind shear and to relate it to the moisture suppression from the Amazon discussed in comment 14. The revised manuscript formulates better this section.

32) Page 7, lines 18-20: "We observe at the same time an intensification of previous convergence zones around complex orography; with a propagation of the convergence areas from the previous

zones towards each other (Fig. 5e)." The magnitude and noisy character of the convergence-divergence patterns indicate that the "convergence zones" are all contaminated by ongoing deep convection in the simulation. So we are basically looking at the environment modified by the storms themselves instead of convergence as a pre- storm lifting mechanism. So I think the above analysis is a bit confusing regarding what exactly the authors wish to discuss. Terrain effects? Lake-induced circulations? The problem is that at this stage (Figs.5e-f) the mesoscale enviroment is already highly modified by the ongoing convection such that it is difficult to isolate the mesoscale forcing mechanisms based on divergence.

**- R:** The referee is right about the later divergence contamination that makes hard to isolate the forcing mechanisms. We have included additional discussion about these features

33) Page 7, lines 31-32: In the analysis of the skew-T diagrams I think the authors missed a few important features: the temperatures and dew-point temperatures are very low if we consider a typical environment for severe convective storms. Naturally, this is explained by the very high elevation of the local terrain, but, still, this point deserves to be stressed since most readers are not familiar with such environments. On the one hand this aspect does not preclude the generation of CAPE (as shown in Fig.6e), but on the other hand it does reduce significantly the precipitable water, which is not informed in the text. In fact, the simulated 2m-specific humidity was rather low over La Paz (Figs.5a-c). Given these points, it is intriguing that a significant flash flood was observed in La Paz, as confirmed by the videos. So this raises a few questions:

has the rainfall alone accounted for the observed flooding or has the melting of hailstones contributed to that occurrence? Or else, is there any indication that the WRF simulation has underestimated the moisture supply for that region? Perhaps, very steep terrain leading to fast surface runoff also accounted for that event? This is an important result and discussion because most forecasters (at least the ones working in lower terrain areas) probably would not cite flash flood as a main threat if looking at those simulated/forecast soundings.

I probably would not say that the thermodynamic profile in Fig.6a is stable (line 31), but approximately moist neutral. It would not be hard to become unstable even with just some little surface heating. By the way, did WRF underestimate the 2m-temperature?

**- R:** We acknowledge the importance of a low freezing level and we discuss it later in the manuscript. However, as stated in the reply to comment 30, we agree this has to be stressed more and we have modified the manuscript to reflect this. The referee also raises a very interesting point regarding the intensity of the rainfall and flash-flood occurrence. To our knowledge, as discussed by Hardy (2009), a mix of elements favored the flash flood (soil saturation by previous precipitation episodes, very steep terrain, and urban characteristics). The hail cumulation over the city center played a role in blocking the sewage system and directing the flows over the streets. We have added a small discussion about this. We also updated the interpretation of Fig.6. And finally, WRF underestimated the 2m temperature but not by much (15 C in WRF and 18 C in La Paz city)

34) Page 8, lines 6 and 8-9: "The location of this band overlaps the lake-valley breeze convergence zone." & "...responsible lifting mechanisms identified until now are orography and lake-valley breeze convergence." As mentioned earlier, I think that a closer analysis of the simulation suggests that the (simulated) lake breeze was enhanced by convectively-induced outflow from previous convection.

**- R:** We present a more insightful analysis following our reply to comment 30)

35) Page 8, lines 6-8: "The evolution of the intensity of vertical velocity at 4000 meters above ground level (magl) and wind shear from surface to 6000 magl (Fig. 7a-c) gives an idea about the severity of afternoon convection and resulting storm." The authors do not develop the discussion here. Were the simulated vertical velocities intense? They do not appear particularly intense to me. Even at 2 km horizontal grid spacing, simulated severe storms can develop vertical velocities of the order of 10 ms-1. Isn´t it possible that WRF has underestimated storm severity in this case? Or maybe it is because large hail accumulation at very high terrain does not require strong updrafts typical of true supercells. Any information available regarding the observed size of the hailstones? Sometimes, at higher elevations, storms produce copious amounts of small hailstones, for which very strong updrafts are not required.

**- R:** The referee is right that we could improve the discussion here. Our results suggest that WRF underestimated the intensity of the event. However, because it is hard to quantify by how much, given the observations limitations, we didn't develop more on this subject. The lack of studies of

this nature on the region makes also difficult to relate our values to similar events. However, we think a small discussion can be fruitful and we have added it.

36) Section 3.3: I think that the discussion around the results from the sensitivy analysis was rather superficial and, therefore, mostly inconclusive.
**- R:** As we have realized in previous sections, we agree our discussion can be improved and we hope the revised version reflects it.

37) Page 10, line 3: "On 19 February 2002, surface wind over the altiplano was guided by thermal lake, mountain and valley breeze effects." Again, I would say that the WRF control simulation suggests that outflow from previous convection also played a role by influencing the strength of the lake breeze (i.e., by enhancing it).
**- R:** We have revised this sentence following our reply to comments 30 and 34

38) Page 10, lines 17-18: "The presence of sufficient wind shear extends and supports the organization of convective storms in terms of multicells, supercells or mesoscale convective systems." In other words, no conclusion was reached regarding the convective mode. Unfortunately, the authors did not check for the possible development (or non-development) of mesocyclones within the storm cells simulated in their D4 domain. That would help identifying the convective mode (as for multicells versus supercells).
- **R:** We have addressed this issue in our reply to comment 13. We have also updated this sentence following the supplementary figure of storm relative helicity. It indeed shows several cells spatially consistent with the updrafts regions shown in Fig.7, but with values not high enough to be considered super-cells (more than 150 m^2 s^-2, according to weather.gov). Further comments on this finding has been added.

[Figure]

39) Page 11, lines 5-6: "....suggests that this severe event was in fact part of a mesoscale convective system." Throughout the text the authors have not presented arguments to sustain such conclusion.
**- R:** We thank the referee for the very useful comments that we take into account and we hope the revised manuscript will be able to present better arguments to sustain our main conclusions.

References:

Hardy S. 2009. Granizada e inundación del 19 de febrero de 2002. Un modelo de crisis para la aglomeración de La Paz. Bulletin de l'Institut français d'études andines:501–514.

Sicart JE, Espinoza JC, Quéno L, Medina M. 2015. Radiative properties of clouds over a tropical Bolivian glacier: seasonal variations and relationship with regional atmospheric circulation. International Journal of Climatology.

Sulca, J., Vuille, M., Silva, Y. and Takahashi, K., 2016. Teleconnections between the Peruvian central Andes and northeast Brazil during extreme rainfall events in austral summer. Journal of Hydrometeorology, 17(2), pp.499-515.

---

## Author Comment (AC2) · 15 May 2019

**Reply to Review from Referee 2 of nhess-2019-27 "Characteristics of a Hailstorm over the Andean La Paz Valley" by Marcelo Zamuriano et al.**

We thank the referee for the fruitful comments and suggestions to improve the quality of the manuscript. We take them gladly into account and we have enumerated them followed by our replies marked with R:

Major comments
1) First of all, the authors must clear that the insights found on the mechanisms underlying the development of this storm are indeed insights on how the WRF simulations works, more than on how this specific storm really evolved. In fact, the observed data are so few, that while the authors can state that the control run has a general agreement with the low resolution satellite-derived observations, they can't prove that the local-scale features evolved follow- ing the details simulated by WRF, since there are no radar data, no lightning data and no hail data to compare with. Also, there is no clear comparison with the only available precipitation data, that is, "The most important rainfall quantity was therefore registered around La Paz next to the mountain slopes (with measured values of around 50 mm)." I assume that 50 mm of precipitation is the daily accumulated value, while the simulations data of Figure 10 show only 3-hours accumulated rainfall, with maximum peaks of about 20 mm. That is not a good forecast if what reported by Hardy (2009) is true, that is, that there were 39.4 mm in only one hour.
**-R:** We understand the referee's concern about the observations limitations that are not able to validate the main features simulated by WRF. We therefore have clarified the numerical nature of the mechanisms by modifying the title to "Numerical Insights of a Severe Convective Storm accompanied by Hail and Flash-flooding over the Andean La Paz Valley" (that can be still modified in our final version). We also have realized during out reply to Referee 1 that our rain-gauge precipitation analysis was rather superficial and we have updated our manuscript and figures to better reflect our findings.

2) While there is much emphasis on hail, from the title of the paper to the tentatives "to investigate the physical processes leading to the hailstorm formation", in the presented material there is really no specific information on hail and in particular on what would favor a severe hailstorm instead of, for instance, a severe storm not characterized by much hail (that is what one would expected from the analysis of only one case study). The Hailcast moduleis coupled into WRF, but I can't find anywhere any clear Hailcast result, like, for example, a map of simulated maximum hailstone diameters. Only a small blue line (very hard to note) is overlapped in Figures 4 and 10, to show hailstone "around 5 mm" or "between 4 and 6 [mm]" of diameters, which seem too few for evaluating the hailcast forecast. Lastly, how important was the hail aspect for the damaged produced by this storm and, in particular, for the 69 casualties? I suspect that the flood aspect (with saturated terrains and steep slopes) could have been more important. In conclusion, maybe that emphasis should be given more on the flash-flood aspects than on hail (and the title changed). Otherwise, the key role of the hail should be better highlighted.
**-R:** The referee is correct that the flash-flooding was responsible for the high number of casualties. We understand the relevance of the flash flood and our updated title takes into account this. We also realize that the hail role in the flash flood is not very clear from our manuscript. As stated by Hardy (2009) the hailstones played a role in blocking the sewage system and directed the flows over the streets; hence the crucial importance of hailstones reproduction by WRF. We acknowledge the referee's point about HAILCAST and the hailstone size that are rather small for hail forecast evaluation. We suspect that WRF underestimated the storm severity and we add some comments about the possible reasons for that, accompanied by clearer figures and captions.

3) The material is not presented in a very linear form and there are many repetitions, like for example between Section 4 Discussion and section 5 Summary and Conclusions. I encour- age the authors to remove most repeated material and to merge together sections 4 and 5.
**-R:** We have realized that our discussion section can be improved and we may have jumped into conclusions without presenting stronger arguments. We gladly accept the referee's suggestion to revise and remove repeated material; at the same time we have updated this section in order to present better our findings.

4) Figures are really too small and very difficult to understand. I suggest to put only one "large" figure per page. Please add at least one map of hail simulated by Hailcast. Because of these major comments, I kindly ask for a major revision.

**-R:** We take into account this comment and we have updated the figures to make them larger and easier to understand.

Minor comments

5) Abstract: The iconic hailstorm "iconic" is an appropriate term? I would remove it here and during the whole manuscript.

**-R:** Perhaps this term is not appropriate, we decided to use rather the term "historical" or just remove it along the manuscript.

6) 1, 19: taking into consideration Bolivian farmers perception of extreme events. Is it pertinent?

**-R:** This sentence points the lack of systematic hail observation system and most of the knowledge comes from farmer perception. We have rephrased this sentence.

7) 1, 20: Andean farmers perceive an increase of the frequency and intensity of storms and hail. How much scientific can be such statement?

**-R:** Local traditional knowledge is still widely used by farmers. While it can not be considered very "scientific", it has been shown that it can be complemented with scientific information for a better communication with local people in rural areas of the Central Andes (Rosas et al. 2016). We point this issue and make clearer our goals in the revised manuscript.

8) 1, 25: was described as an unprecedented crisis. Maybe crisis is not the best word to use here.

**-R:** We have changed it to "emergency"

9) 2, 1: (that resulted in 69 casualties How they died? Because of flood, wind gust, giant hail?

**-R:** In conformity to our reply to comment 2, we have updated this sentence to make clear the flash-flood was the main driver for casualties.

10) 2, 5: the knowledge is taken from local peoples perception of hail frequency and intensity. That is too few to write a paper on "hailstorm".

**-R:** The goal of our original sentence is to highlight the observational limitations but at the same time the urgency of a better understanding of this kind of events over the region. We have better justified the motivations and updated the title.

11) 2, 5-7: The lack of formal physical process knowledge about local thunderstorms formation over this region is evident as we take as example the explanation given by the SENAMHI about the plausible mechanisms for this particular cell formation. I don't think that criticizing SENAMHI explanation is relevant here. If their explanation "might sound trivial for a super-cell formation", also your explanation that convective cells "were triggered by a mix of low level wind convergence, surface heating and orographic forcing" (page 10, line 8) might sound trivial to others.

**-R:** We can see that the term "trivial" might not be the best one here. We have reformulated this sentence.

12) Introduction: In this section I would expect some explanation about literature studying hailstorms cases, like for example the work by Kunz et al. (QJ RMS 2016), if emphasis on hail will be maintained. Also some references to orographic precipitation studies could be added, like for example seminal works from Rich Rotunno, Robert Houze, Daniel Kirshbaum and others.

**-R:** We are happy to expand the introduction with more hailstorm cases in other regions. This is also useful for later on when we discuss how our case study can be compared with other similar studies in other regions.s.

13) 3, 4: 1745 LST: I searched in Internet for what means LST and it is the "Local Sidereal Time", while you probably was reffering to the "Local Time" (LT). Please correct all LST and describe the relation between LT and UTC time for your specific location.

**-R:** The local solar time (LST) is defined in page 1, line 24. We decided to use it since we discuss daytime thermal circulation and we use such terms as "late morning" and "early afternoon" along the manuscript. However, we understand the referee concern about not using UTC time and we added the LST - UTC times correspondence (LST = -4 UTC) in our revisions.

14) Figure 3a: I can't really see the observed rainfall distribution. Maybe you can plot the accumulated values at each SENAMHI station located in that area?
**-R:** We realize many figures may not be well seen and we have updated them.

15) 4, 16: We study the role
**-R:** Thank you, we have used this correction.

16) 4, 21: We assess the presence of the main ingredients for a hailstorm to occur (moisture, instability and lifting): First of all, that language is more appropriate for a cookbook recipe than for a scientific article. Second, if there are -unfortunately- some people that oversimplify the thunderstorm forecasting problem to such a level, they do it for thunderstorm in general, not specifically for "hailstorm". So, please, describe some general features of the environmental conditions that favor thunderstorm development and, if you have any evidence, of specific conditions that instead favor hailstorm formation. Otherwise, simply list the parameters that this study investigates.
**-R:** Both point are fair. We have rephrased this paragraph in order to convey better the methods used and main findings.

17) 4, 23-24: The low level moisture transport vectors were calculated following IVTU = 1  200 qudp
 g SFC . . . It is calculated from the surface SFC up to 200 hPa.
I really can't understand how can you call it "low level" moisture transport if it is calculated up to 200 hPa!
**-R:** The referee is correct, the calculation is made for all the column up to 200 Hpa. We have removed "low level".

18) Figure 2a-2d: Please define what are "sensor count"?
**-R:** We have added the definition.

19) 4, 27-28: the presence of low level water vapor is not well captured in this band but its corroborated with infra-red image at 12μm (not shown).
Why infra-red at 12μm should provide information on low level water vapor?
**-R:** The referee is right that this band is not useful to detect low level water vapour, but rather surface temperature and moisture. Since the results are not much different to the water vapour band (6.5 μm channel) we opted to use the 12 μm channel to explore  the soil moisture and saturation that could have played a role in the flash-flood. We have added figures from both bands (water vapour and infrarred) to the appendix and rewritten this paragraph in order make it clearer. We keep the visible band results in the manuscript's improved Fig.2.

20) 6, 8-9: We note that the models rainfall spatial distribution corresponds very well to the clouds locations in Fig. 2a-b). While there is a general agreement, I would not say that forecast and observation fits "very well". Can you provide any verification measure of the agreement between observations and forecasts?
**-R:** We hope the updated Fig. 2 will be able to show the agreement. A comment about verification measures was also added.

21) 6, 14-15: Thus WRF is able to simulate the event with its most important features. Not so sure, also because you do not have enough observations to describe in detail the event fea- tures.
**-R:** We have reformulated this sentence with the updated figures.

22) 6, 20: (with measured values of around 50 mm).
In 24 hours? What is the WRF forecast in the same period and location? Is 50 mm in one day an exceptional rain value? BTW, what is the rain climatology for that location and period of the year?

-R: We realize our precipitation analyse from raingauges was rather superficial. We have improved Fig. 3a and added information about rain climatology.

23) 6, 24: The analysis of the large scale characteristics and the few observations available provides insufficient information about the three basic ingredients for a thunderstorm: moisture, instability and lifting.
Maybe it is better to remove it, since it does not add any useful information.
-R: We have replaced this sentence.

24) 6, 28: in order to explore the chronology of the precipitation. "chronology" is appropriate?
-R: Perhaps "evolution" is more appropriate. We have changed this.

25) 7, 3: A closer look to the maximum radar reflectivity. A closer look to the simulated maximum radar reflectivity.
-R: Yes. Thank you for pointing this out.

26) 7, 6: even hailstones of around 5 mm are simulated at the centre of the two formed cells. Is it the Hailcast maximum diameter? Or the mean diameter? How it compares with observed hailstone diameters (e.g. from media report) and with the locations were hail was reported? Why not showing also a full map of hail diameters as simulated by Hailcast?
-R: This information is indeed missing. HAILCAST provides the mean hailstones diameter on surface alongside with their standard deviation. The hailstones diameters over this region are relatively small (personal communication with SENAMHI). We have added a sentence about how simulated and observed hailstones sizes compare.

27) 7, 11: comes from the Amazon avoiding the cordillera obstacle "avoiding" is the appropriate term?
Figure 5a-c: Reference vectors are 50 [kg m−1s−1] ?
Figure 5d-f: Reference vectors are 3 [ms−1]?
-R: We have replaced the term and updated the figure captions.

28) 7, 29: in the proximity if the rain-band. in the proximity of the rain-band
-R: Corrected now.

29) 8, 2: the atmosphere is saturated until 400 hPa with important wind shear favoring hail and graupel formation. Can you add a reference to support the hypothesis that wind shear favor hail and graupel formation? In figure 6f I can see a strong directional shear at about 500 hPa, but for example also in figure 6e there is a strong directional shear at about 550 hPa, even if wind directions are completely differ- ent. So, the important issue was really the wind shear or rather the wind direction and intensity absolute values?
-R: This comment is pertinent and reinforces the need of an extended literature review in other regions. We have added further references to the introduction section and used it here to argument the updated discussion.

30) 8, 7: and wind shear from surface to 6000 magl (Fig. 7a-c)
How you define wind shear? Is it the "bulk shear"? I.e. taking the magnitude of the vectorial difference between wind at 6 km and wind at 10 m? Please explain.
-R: It is indeed the vectorial difference between wind at 6km and 10 m. The definition is now included.

31) 8, 13: Nevertheless, convergence is not enough to explain deep convection. Is that a scientific explanation?
9, 9: showing that breeze and orographic lifting are enough for producing rainfall. Is that a scientific explanation?
9, 17: leaving the cordillera storm free with isolated hailstorms. Sorry, can you rephrase?
10, 6: following the thermo-topographic circulation. Can you explain better?
10, 13-14: This propagation allowed both cells to join each other resulting in a precipitation band. This auto-propagation mechanism has been observed Please, can you explain better?

-R: We recognize the referee's concerns in this section. We may have been overly confident in our figures and we realize a stronger argumentation has to be made. We hope the revisions made to this section will meet the journal standards and better convey our findings to the readers.

32) 10, 17-21: The presence of sufficient wind shear. . . as shown in Fig. 7f.
This part seems not much relevant with the experiments described. I suggest to remove it or to move it to Section 3.
-R: We have moved this part to section 3.

37) 11, 5-6: satellite information and reanalysis suggests that this severe event was in fact part of a mesoscale convective system. That option was never analyzed nor mentioned before the Conclusions. Please remove it or discuss it with supporting facts already in the previous sections.
-R: The referee is correct that the mesoscale convective system nature was not analysed and we have accepted his advice to remove it.

38) 11, 20-21: And the surface heat flux suppression (NOHEAT) highlights the importance of surface energy fluxes for atmospheric instability. Isn't it a trivial result?
-R: It is indeed trivial, and doesn't highlight the main result that even without heat fluxes, WRF is able to simulate convection by orographic influences (Fig. 10 d). We have modified this part.

39) 11, 22-23: highlights the complex interaction between large scale circulation, orography and local features in the formation of hailstorms over the tropical Altiplano. Sincerely, I have not found any specific information that can explain why a hailstorm was formed, instead than a thunderstorm (or supercell) not particularly characterized by hail.
-R: This is true and we have therefore modified the title and the text within this manuscript.

40) 11, 23-24: A semi-comprehensive scheme of participating mechanisms can be found in Fig. 11. Please, explain how figure 11 describes this (participating?) mechanisms in details or remove this figure.
-R: We have added a description of figure 11.

41) 11, 29-30: the proposed mechanisms of this hailstorm formation should be confirmed by high resolution observations. You already said that these observations are not available.
-R: We concede observations are not currently available and our intention was to stress the need of an appropriate observation system that would be able to confirm our findings. Maybe not in the near future, but we hope this study will stimulate decision makers to take a step towards a modernization of the Bolivian weather observation system.

REFERENCES

Rosas, G., Gubler, S., Oria, C., Acuña, D., Ávalos, G., Begert, M., Castillo, E., Croci-Maspoli, M., Cubas, F., Dapozzo, M. and Díaz, A., 2016. Towards implementing climate services in Peru–The project CLIMANDES. Climate Services, 4, pp.30-41.

---

## Author Comment (AC3) · 15 May 2019

**Reply to Referee 3 in respect to his Review „Characteristics of a hailstorm over the Andean La Paz Valley" by Marcelo Zamuriano et al.**

The manuscript examines a severe convective storm event associated with large hail and heavy rainfall that occurred in February 2002 over the La Paz region in South America. Due to the scarce availability of suitable observational data, WRF simulations complemented by sensitivity studies with different model setups were conducted. The main objective of the study is to identify the mechanisms and processes most relevant for the triggering and maintenance of the convective storms. The topic of the paper is basically relevant for NHESS. However, there are a number of important issues that have to be considered and addressed before the paper can be published. Above all, an in-depth discussion of all figures, the findings and the most important results is necessary. All my suggestions and comments are listed below as major, minor, and (a few) editorial points.

We are thankful to the referee for his/her valuable comments and suggestions. We have gladly taken this review into account to improve the manuscript. The referee's points are followed by our replies starting with an R:

Major revision points:
1) The results and illustrations are mostly discussed very briefly, superficially and descriptively. A thorough, in-depth discussion and interpretation is lacking. This applies in particular to the sensitivity studies and to the application of the Hailcast model. In order to use them in a sensible way, a much deeper discussion and interpretation of the results is necessary. Furthermore, the interconnection of all subsections has to be improved.
**-R:** After having re-read our original manuscript, we are aware that our discussion section can be vastly improved and solid arguments are still lacking. We hope our revised version can overcome this issue in particular with respect to the observations from rain-gauges, the HAILCAST model, the discussion section and sensitivity studies.

2) Please explain in which way the result can be generalized and how the findings may apply to comparable events.
**-R:** We have added this part to the conclusions section.

3) I do not agree that the model capture well the main features of the severe thunderstorms. There are large discrepancies in both the location and precipitation intensity. Even the two major convective cells are not well reproduced by the model. The conclusion (Sect. 3.1.2) that the model is able to simulate the main features and, thus, capture the mechanisms decisive for the triggering and maintenance of the storms is not justified.
**-R:** We agree with the referee that our arguments in the original manuscript are not very well elaborated. Further analysis and a deeper look to rain-gauges observations even suggest that WRF underestimated the intensity of the event. However, because it is hard to quantify by how much given the observations limitations we didn't develop more. The lack of studies of this nature on the region makes also hard to contrast our values with similar events. We have updated the manuscript in order to convey better our main findings.

4) The effect of a lake breeze that is relevant for convection triggering cannot be derived solely from the wind field, but requires analyses of temperature / moisture gradient or vertical lifting. As sea/lake breezes have a limited vertical extent of a few hundred meters, they cannot be detected at 500 hPa, not even over the elevated terrain around Lake Titicaca.
**-R:** We realize we have mixed lake breeze (using surface wind) and 500hPa wind circulation without explaining the reasoning behind it. The main point of our original sentence was to relate surface to low level circulation (500 hPa) for the purpose of introducing wind shear and to relate it to the moisture suppression from the Amazon discussed during the synoptic analysis. We have conducted revisions to elaborate better this section.

5) The discussion Section (together with the Conclusion Section) is more a summary than really a discussion (this applies at least for the first half of the text). I'd like to see a more thorough discussion and a better synthesis of the results instead of repeating what was already written. I also suggest to extend the comparison with other related studies and references.

**-R:** As stated in our reply to comment 3, it is hard to contrast with similar studies in the central Andes because of the lack of them. Nevertheless, our findings can be contrasted with similar studies in other regions. We have added a literature review about similar events in other regios to our introduction and updated the discussion section accordingly. We hope the revised manuscript reflect the improvements with better synthesis of the results.

6) Be accurate in your citations!
Sect. 2.1.2: I'm puzzled about the statement "known uncertainties of precipitation estimates over complex terrain (Rasmussen et al., 2013)" as this does not make sense for the TRMM algorithm. The cited reference investigated the range of the rain bias in storms containing four different types of convection in extreme radar echoes over South America, but did not investigate a relation between bias and terrain characteristics. The study of Kunz et al., 2018 investigates a supercell over low-mountain ranges and not over the Alps.
These are just two examples that I've checked.

**-R:** We have taken more care to our citations in the revised manuscript.

7) The number of cited references is rather small.
**-R:** As discussed in our reply to comment 5, we have expanded our literature revision and added further references. Also the discussion section expands the references used.

8) Most of the Figures are too small and/or have a too low resolution; it's hardly possible to see any details. Furthermore, as most of the readers are not familiar with the study area, anadditional figure that indicates all areas and cities referenced in the text (Altiplano, Amazonas, ...) would be helpful (Fig. 3a is too small!).
**-R:** This was a recurrent problem for all referees. We agree that the figures can be hard to see and we have updated them in our revised manuscript.

9) The English language requires a thorough check by a native speaker. I have listed a few corrections, but these are too numerous to fully list.
**-R:** The referee is correct to point out the English errors. We are following your advise and making the revised manuscript checked by a native speaker.

Minor points
1) The title should be changed as the paper goes beyond the investigation of one single hailstorm.
**-R:** We have considered that a title that better summarize the outcome of the research would be: "Numerical Insights of a Severe Convective Storm accompanied by Hail and Flash-flooding over the Andean La Paz Valley"

2) Abstract: Can you highlight the most important findings (e.g., We show the importance of orographic configuration... is too general; be more specific). Consider to move the last sentence after the 2nd one.
**-R:** We agree the abstract is too general and it has been updated to include more specific findings.

3) The use of LST is weird: Twelve noon local solar time (LST) is defined as when the sun is highest in the sky, i.e., it is a function of the geographical coordinates. Do you mean local time instead (LT)? What is LT in UTC at La Paz?
**-R:** Since we use official local times, we have replaced LST by LT. We have also introduced the equivalence between LT and UTC (LT = -4 UTC) in the manuscript.

4) Introduction, 1st paragraph: Give a brief reason why hail hazard assessment is not available. But also note that there are some studies available that uses either satellite data (e.g., Cecil and Blankenship 2012) or reanalysis (e.g., Prein and Holland 2018) to estimate hail frequency, that may be mentioned (e.g., by giving an estimate how frequent the region is affected by SCSs, similar events in the past, ...).

**-R:** We agree this needs further development and we have pointed out the observation limitations over the region in the updated introduction. We have additionally included further literature review in order to explore how this event can compare to similar ones in other regions.

5) Introduction, 2nd paragraph: give some more details about the event: total rainfall accumulation, hail sizes, wind gusts; what was the reason for the large number of casualties?
**-R:** We have included the details of rainfall accumulation. We also have expanded the introduction including the possible effects from hailstones over the flash flood that was responsible for the casualties. As stated by Hardy (2009) the hailstones played a role in blocking the sewage system and directed the flows over the streets); hence the crucial importance of hailstones reproduction by WRF. We have made sure to include this information in our manuscript.

6) P2L5: "The lack of formal physical process knowledge" is unclear: I think you mean what makes this region special related to thunderstorms compared to others? The physical processes for thunderstorm formation are the same all over the world.
**-R:** The referee is correct. This sentence has been rephrased.

7) Introduction, last paragraph: Say a few more words about the motivation and objectives of this study to be published. Only saying the goal is to better understand the processes is too simple. What are the research questions / hypotheses? What are the reasons for performing sensitivity studies?
**-R:** We have added a paragraph to the motivations and objectives, including the reasons for performing sensitivity studies.

8) P3L14: how many gauges? If all stations are shown in Fig. 3a, you cannot say "around 1 km distance" given the large spatial differences.
**-R:** We have expanded the stations network description alongside with the station network climatology in order to assess the severity of this event.

9) P3L16: "some data quality issues": what do you mean?
-**R:** We have added some examples.

10) Sect. 2.2.1, 2nd paragraph: give some more details about the schemes and the configurations you have used (besides: write out YSU); also give a reference for the radar-forward operator you have used to create Fig. 4. Furthermore, you should motivate here - or maybe better in the introduction (see minor point 7)- the reasons why you have conducted the sensitivity experiments. What do you expect from those additional runs?
**-R:** We have modified the introduction in order to contain the sensitivity studies motivations, and we have added more details to the description of the schemes and configuration used.

11) 2.2.2: Why did you not consider deep layer shear (DLS) or storm-relative helicity SRH, which are important ingredients for supercells? The unit of IVTU/IVTV in 100 g (s m)-1 (yes, this is the result when using the units you stated) is very strange and not an SI unit. I strongly recommend to insert dp not in hPa, but in Pa. Why do you integrate until 200 hPa?
-**R:** The referee is correct. We thank you four this remark. Specific humidity units must be in kg kg^-1 and dp units should be in Pa, we have corrected this. We haven't used the SRH in our manuscript since the main goal was not to asses the supercellular nature of the event. However, and in agreement with the other referees, we accept it would be useful to discuss this parameter in order to compare our case event to storms in other regions. We have added a small discussion regarding SRH in the revised manuscript. Finally, we decided to update our IVT formula until 50 hPa since the integration should be made over the full column; the original calculations are not much different to the updated calculations since the Bolivian High position (Fig. 1a) regulates the upper levels flow coming from the west and this flow is dry in general.

12) Always put a blank between two units (e.g., ms-1 could be m/s or millisecond).
-**R:** Now is corrected.

13) How is CAPE computed? Mixed layer CAPE, most unstable CAPE, ...?
**-R:** We refer to the most unstable cape. We added this information.

14) P5L4-6: As you did not investigate runoff or flash floods, soil saturation is irrelevant. You may mention this in the introduction, but not in the result section.
-R: the referee is correct. We have moved this sentence to the introduction.

15) Sect. 3.1.1 / Fig. 1b: Showing only one isoline of the geopotential at 200 hPa (also without labelling) does not make sense as the flow at lower levels, which are most important for the triggering, could be completely different from that at 200 hPa.
-R: The 200 hPa level geopotential is generally used as proxy for moisture transport over the Altiplano (Garreau, 2003). We agree that a measure of the intensity (by labelling it) is lacking since its position and intensity results on the enhancement (southern position) or suppression (northern position) of moisture transport towards the Altiplano. We acknowledge that a configuration favoring the moisture was not shown and we decided to add such configuration in Fig. 1a.

16) P5L13-14: what do you mean by "northward displacement of the Bolivian High"?
-R: We have added the normal summer configuration of the Bolivian High in Fig. 1a (favoring moisture transport). We have modified this sentence in accordance to the figure update.

17) P5L18: specify what is meant by "mesoscale features"
-R: We added a sentence specifying such features.

18) Fig. 2: There is a discrepancy between blue shaded contours and the color bar at the bottom (besides: the numbers are illegible).
-R: Fig. 2 has been updated.

19) P5L24: "...show a remarkable spatial consistency..." this statement is too general and too optimistic; of course, there is some consistency (mainly at 17 LST), but also some discrepancies. This is what one would expect as you compare visible cloud areas with rainfall.
-R: We agree we may have been too optimistic here. We have rephrased this sentence in accordance to the updated Fig. 2

20) P5L25: "Titicaca lake, the Amazon region, and the eastern cordillera..." These regions / features should be indicated in one Figure.
-R: These features are indicated in Fig. 3a. However, we concede it is hard to see and we therefore have moved and updated it into Fig. 1.

21) P5L28-30: "important convection...shallow convection". From the visible channel solely, you cannot distinguish the intensity of convection; "TRIMM is not able to capture any light rainfall". Why do you suppose that rainfall already started at that time?
-R: We realize Fig. 2 alone do not provide enough evidence for sustaining these claims. So we decided to rewrite this sentence with a newer version of Fig. 2

22) P5L7: Why do you consider longwave radiation here? Explain and motivate this (and recall that Figs. 2a-d show the visible channel).
-R: Past studies (Sicart et al. 2015 and Sulca et al. 2018, among others) have shown the usability of OLR (outgoing longwave radiation) for cloudiness detection over this region. However, its usability is limited since it contains information about most longwave spectrum and it's hard to discriminate low level vapor to cloudiness, especially during the morning. We discuss this part a bit more in the revisions.

23) P5L10: The Figures show radiation and precipitation, and not water vapor as stated here.
-R: The referee is right. We have rewritten this sentence.

24) P6L25: Also consider vertical wind shear.
-R: We have considered it in the results section and we have revised the methods section to make them better interconnected.

25) Sect. 3.2.1, 1st paragraph: So what? What can you conclude from the Hovmoeller plot?
-R: We present more arguments before jumping to conclusions in our revised manuscript.

26) Sect. 3.2.1, 2st paragraph (see also major comment 4): The strong north-westerly flow east of the lake between 11 and 13 LST can also originate from the cold pool of the convective cell. This may also explain the large flow divergence at 13:00 around that cell. Note that sea/lake breezes have a rather small vertical extent so that the associated wind field is of minor importance for the movement of the convective cells.
**-R:** The referee makes an interesting statement about the enhancement of the lake breeze by early convective activity over the lake. We have included further comments on this feature on the revisions.

27) Fig. 5a-c: the color code of the figures and the color bar do not match; what are the areas indicated by the brown color?
**-R:** Brown area is orography. We have replaced it with dashed contour lines indicating the 4000m orography line.

28) P7L10: "While the surface humidity follows the lake breeze..." I cannot see any relation.
**-R:** We have modified the figures and the text accordingly to better argument the relationship between lake breeze and humidity transport.

29) Sect. 3.2.2, 2nd paragraph (see comment 26): sea / lake breezes typically have a vertical extent of a few hundred meters. They cannot be identified at 500 hPa. Even though at 14:00 there seems to be a frontal boundary involved, which was not the case at 11:00 and even at 12:30. In the previous paragraph, however, you suggested the wind field even at 11 and 12 is associated with the lake breeze. Furthermore, I'd suggest not using different times for the plots (12:30 in Figs. 5-6 is not shown by Fig. 4). Finally, relate the fields to the convective systems.
**-R:** Major comment 4 also addresses this misunderstanding. We realize we have mixed lake breeze (using surface wind) and 500hPa wind circulation without explaining well the reason within. So we have rewritten this part to make ir clearer. We also recognize the difference of times for Figs. 5-6 not shown in Fig. 4 and we have decided to modify Fig. 4 to make them compatible. In addition, we added some comments between the fields and convective systems relationship.

30) Fig. 6: It's hard to see the CAPE. I suggest to include additional Figures showing only CAPE. Furthermore, it's not clear what Figs. 6a-c show.
**-R:** We have modified Fig. 6a-c and the text accordingly.

31) P7L24-26: It's rather confusing to term that sensible heat is released as the Figures show the vertical heat flux; Solar radiation is surely not the only reason for increases in CAPE (cf: the largest increase in CAPE in the northeast is associated with lowest sensible heat flux.)
**-R:** We agree a deeper discussion on sensible heat and CAPE is needed here and we have subsequently modified this part.

32) P7L30: As the city heat-island effect is not relevant for the convective storms, I suggest to omit this statement.
**-R:** We have rewritten this statement.

33) Fig. 7/8: It would be easier for the reader to show shear as amount and not as vector.
**-R:** We have modified Fig. 7/8 to show shear in a more readable manner.

34) Sect. 3.2.4: In some places reference is made to convection triggering. However, convection on that day is triggered earlier as shown in Fig. 4. Be sure what is really meant here: maintenance, triggering of new cells, or – as you describe – the merging of scattered convection to a larger band showing some basic features of a squall line.
**-R:** We have followed the referee's suggestion and modified the text correspondingly.

35) P8L12-14: Explain where the cold pool is located (I cannot see a convergence line; besides, the cold pool cannot be situated over the convergence line). It's not possible to speculate about a supercooled state without showing hydrometeors.

**-R:** We have modified Figs. 7/9 to point the cold pool locations and we have added further discussion. the referee is right about the speculation about a supercooled state and we decided to modify this to "cold atmospheric environment" (above the freezing level)

36) P8L21: This explanation for a cold pool is wrong; rather evaporation and sublimation cooling by hydrometeors drives the cold pool.
**-R:** We have corrected this sentence

37) P8L28-29: Literature?
**-R:** We have based this claim on Fig. 8a-c. But we realize this sentence can be expanded with further arguments, which we have made.

38) P8L7: "convection is present without thermodynamic instability" this is a contradiction in itself.
**-R:** We have rephrased it.

39) P9L13: How is hail formation suppressed? Changes in microphysics, intensity/size of the updraft, or what else?
**-R:** We realize the mechanism is missing and we have modified the text to include plausible explanations.

40) Experiment with no lake: when you state that the wind field for this realization is similar to a lake breeze, than I doubt even more that really a lake breeze is responsible for CI.
**-R:** We have updated this section taking into account the additional analysis of lake breeze enhancement by early convective activity.

41) Caption Figure 10: Make clear that the smaller "hailstones" with a diameter of 4 mm are treated as hail in WRF only; according to WMO definition, hail has a minimum diameter of 5 mm.
**-R:** We have adopted your suggestion. We also added a comment about the hailstones diameter standard deviation simulated by WRF (same order as the mean diameter)

42) Discussion: All the above critic points likewise apply to the Discussion Section
**-R:** We accept this comment and we modified the discussion section accordingly.

43) P10L13: Cold pools behave as density currents. Thus, their propagation results from the interaction between mean flow and density current. They do not directly propagate with the mean wind. Furthermore, cold pools emerging between two cells leads to flow divergence, which prevents and not favors cell merging. The term "auto-propagation" is not appropriate here.
**-R:** We agree this term can be misleading in this context and we decided to rephrase the sentence.

44) P10L17-18: Low to moderate wind shear does not allow for substantial hail formation as it also affects the strength / width of the updraft (cf. Dennis and Kumjian, 2017 in JAS).
**-R:** We have included additional discussion about how our findings can be compared to other regions and we have included the wind shear parameter in it.

45) P10L24: The conclusion about a relation between instability and surface fluxes are not justified as the largest increases in CAPE occur in a region with lower fluxes (NE parts).
**-R:** We have expanded this discussion in order to relate CAPE to other parameters.

46) P10L26: The reduced temperature gradient was not shown.
**-R:** This is true. We include the values in our revisions.

47) Section 4, last paragraph: The question of a possible trend in the frequency of hailstorms is irrelevant for a single case study.
**-R:** We have moved this paragraph to the conclusion part in order to motivate further research.

48) Conclusions: The most important results should be more clearly identified and highlighted. Clearly show what is new, what the reader should have learned from the study.

**-R:** We agree with this comment. We hope the updated conclusions communicate better our findings.

49) P11L6: Usually an MCS is defined as an ensemble of thunderstorms that produce a contiguous precipitation area on the order of 100 km or more in horizontal scale in at least one direction (e.g., AMS glossary). This definition does not apply to the 19 Feb. storm.
**-R:** This is correct and we decided to remove the MCS part.

50) P11L6 (blocking): You haven't shown that blocking really occurred on that day. This, of course, would increase the substance of the paper.
**-R:** We agree with the referee in this part. We have consequently added further evidence for blocking in our revised manuscript.

51) P11L23-24: I suggest to move this part to the discussion section, but also to explain in detail Figure 11.
**-R:** We agree that Figure 11 should be detailed within the text and we have added this explanation. It has also been moved to the discussion section.

52) Typos / Small corrections
**-R:** We thank the referee for all the small corrections that we made sure were incorporated in the revisions. We also have checked again the manuscript with the help of a native speaker.

References:

Hardy S. 2009. Granizada e inundación del 19 de febrero de 2002. Un modelo de crisis para la aglomeración de La Paz. Bulletin de l'Institut français d'études andines:501–514.

Sicart JE, Espinoza JC, Quéno L, Medina M. 2015. Radiative properties of clouds over a tropical Bolivian glacier: seasonal variations and relationship with regional atmospheric circulation. International Journal of Climatology.

Sulca, J., Vuille, M., Silva, Y. and Takahashi, K., 2016. Teleconnections between the Peruvian central Andes and northeast Brazil during extreme rainfall events in austral summer. Journal of Hydrometeorology, 17(2), pp.499-515.